# Severe Californian wildfires in November 2018 observed from space: the carbon monoxide perspective

Oliver Schneising, Michael Buchwitz, Maximilian Reuter, Heinrich Bovensmann, and John P. Burrows

Institute of Environmental Physics (IUP), University of Bremen FB1, Bremen, Germany

**Correspondence:** O. Schneising (oliver.schneising@iup.physik.uni-bremen.de)

**Abstract.** Due to proceeding climate change, some regions such as California face rising weather extremes with the dry periods becoming warmer and drier entailing the risk that wildfires and associated air pollution episodes continue to increase. November 2018 has turned into one of the most severe wildfire episodes on record in California with two particularly destructive wildfires spreading concurrently through the North and the South of the state. Both fires ignited at the wildland-urban interface causing many civilian fatalities and forcing the total evacuation of several cities and communities.

Here we demonstrate that the inherent carbon monoxide (CO) emissions of the wildfires and subsequent transport can be observed from space by analysing radiance measurements of the TROPOspheric Monitoring Instrument (TROPOMI) onboard the Sentinel-5 Precursor satellite in the shortwave infrared spectral range. From the determined CO distribution we assess the corresponding air quality burden in Californian major cities caused by the fires and discuss the associated uncertainties. As a result of the prevailing wind conditions, the largest CO load during the first days of the fires is found in Sacramento and San Francisco with city area averages reaching boundary layer concentration anomalies of about $2.5\,\mathrm{mgCO\,m^{-3}}$. Even the most polluted city scenes likely comply with the national ambient air quality standards ($10\,\mathrm{mgCO\,m^{-3}}$ with 8-hour averaging time). This finding based on dense daily recurrent satellite monitoring is consistent with isolated ground-based air quality measurements.

## 1 Introduction

As a consequence of climate change, precipitation and temperature extremes in California during the cool season (October-May) are occurring more frequently with the dry periods becoming warmer and drier (Swain et al., 2016), which is associated with an increased fire risk. The increasing number of people living in the wildland-urban interface paired with proceeding climate change entailing longer lasting and more intense fire seasons temper the outlook for the future (Radeloff et al., 2018).

The wildfire season 2018 has been the most destructive on record with respect to burned land area, destroyed buildings, and fatalities (California Department of Forestry and Fire Protection, 2019). After a series of extensive fires in July/August including the Mendocino Complex, the largest wildfire in California since the beginning of recording, another round of large wildfires erupted in November, most prominently the Camp Fire and the Woolsey Fire. The Camp Fire started in the morning of November 8 in Butte County in the North of the state and grew rapidly. It became both California's most destructive and deadliest wildfire since records began. The Woolsey Fire ignited on the same day as the Camp Fire in the early afternoon near

the boundary between Los Angeles and Ventura counties and burnt all the way to Malibu. Both fires forced the total evacuation of several cities and communities.

Smoke from the fires also reached the major cities of the state prompting health warnings and the advice to remain indoors or wear face masks in certain areas (Sacramento Metropolitan Air Quality Management District, 2018; Bay Area Air Quality Management District, 2018). The air quality was affected by particulate matter and carbon monoxide (CO), which results from the incomplete combustion of biomass during wildfires (Yurganov et al., 2005). CO is a colourless, odorless, and tasteless gas that is toxic in large concentrations because it combines with hemoglobin to carboxyhemoglobin, which cannot effectively transport oxygen anymore. As a consequence, it has the ability to cause severe health problems (Omaye, 2002). CO also plays an important role in tropospheric chemistry being the leading sink of the hydroxyl radical (OH) and acting as a precursor to tropospheric ozone (The Royal Society, 2008).

The Environmental Protection Agency (EPA) is required to set National Ambient Air Quality Standards (NAAQS) for six pollutants considered harmful to public health and the environment, including carbon monoxide, by the Clean Air Act. The CO standards are fixed at $9\,\mathrm{ppm}$ (corresponding to $10\,\mathrm{mg\,m^{-3}}$ for normal temperature and pressure) with an 8-hour averaging time, and $35\,\mathrm{ppm}$ ($40\,\mathrm{mg\,m^{-3}}$) with a 1-hour averaging time, neither to be exceeded more than once per year (U.S. Environmental Protection Agency, 2011).

Several spaceborne instruments have been measuring CO on a global scale including the Atmospheric Infrared Sounder (AIRS) (McMillan et al., 2005), the Tropospheric Emission Spectrometer (TES) (Luo et al., 2015) and the Infrared Atmospheric Sounding Interferometer (IASI) (Clerbaux et al., 2009), which observe emissions in the thermal infrared (TIR) and are mainly sensitive to mid/upper-tropospheric abundances. The sensitivity of TIR satellite sounders to near-surface CO concentrations varies with the thermal contrast conditions (Deeter et al., 2007; Bauduin et al., 2017). The Measurement of Pollution in the Troposphere (MOPITT) instrument (Drummond et al., 2010) combines observations of spectral features in the TIR and in the shortwave infrared (SWIR) to increase surface-level sensitivity in some scenes (Worden et al., 2010). Nearly equal sensitivity to all altitude levels including the boundary layer can be achieved from radiance measurements of reflected solar radiation in the SWIR part of the spectrum. This was first demonstrated by CO retrievals from the SCanning Imaging Absorption spectroMeter for Atmospheric CHartographY (SCIAMACHY) instrument (Burrows et al., 1995; Bovensmann et al., 1999) onboard ENVISAT (Buchwitz et al., 2004; de Laat et al., 2010) in the $2.3\,\mu\mathrm{m}$ spectral range.

Until now, the satellite-based analysis of CO emissions from fires has been utilising profile or column information from, e.g., AIRS (Fu et al., 2018), IASI (Turquety et al., 2009), the Microwave Limb Sounder (MLS) (Field et al., 2016), MOPITT (Deeter et al., 2018), and SCIAMACHY (Buchwitz et al., 2007; Borsdorff et al., 2018b). The recent TROPOMI instrument offers an unique combination of high precision, accuracy, spatiotemporal resolution, boundary layer sensitivity, and global coverage fostering the monitoring of near-ground CO sources (Borsdorff et al., 2018a; Schneising et al., 2019).

## 2 Data and Methods

In this study, we derive and analyse atmospheric carbon monoxide from the radiance measurements of the TROPOspheric Monitoring Instrument (TROPOMI) onboard the Sentinel-5 Precursor (Sentinel-5P) satellite (Veefkind et al., 2012) using the latest version of the Weighting Function Modified DOAS (WFM-DOAS) algorithm (Buchwitz et al., 2006; Schneising et al., 2011) optimised to retrieve vertical columns of carbon monoxide and methane simultaneously (TROPOMI/WFMD v1.2) (Schneising et al., 2019).

Sentinel-5P was launched in October 2017 into a sun-synchronous orbit with an equator crossing time of 13:30. TROPOMI is a spaceborne nadir viewing imaging spectrometer measuring solar radiation reflected by the Earth in a push-broom configuration. It has a swath width of 2600 km on the Earth's surface and covers wavelength bands between the ultraviolet (UV) and the shortwave infrared (SWIR) combining a high spatial resolution with daily global coverage. The horizontal resolution of the TROPOMI nadir measurements, which depends on orbital position and spectral interval, is typically $7 \times 7\,\mathrm{km}^2$ for the SWIR bands used in this study. Due to its wide swath in conjunction with high spatial and temporal resolution, the observations of TROPOMI yield $CO$ amounts and distributions with unprecedented level of detail on a global scale (Borsdorff et al., 2018a).

As a result of the observation of reflected solar radiation in the SWIR part of the solar spectrum, TROPOMI yields atmospheric carbon monoxide measurements with high sensitivity to all altitude levels including the planetary boundary layer and is thus well suited to study emissions from fires. In order to convert the retrieved columns into mole fractions, they are divided by the corresponding dry air columns obtained from the European Centre for Medium-Range Weather Forecasts (ECMWF) analysis. Thereby, the ECMWF dry columns are corrected for the actual surface elevation of the individual TROPOMI measurements as determined from the Global Multi-resolution Terrain Elevation Data 2010 (GMTED2010, United States Geological Survey (2018)) inheriting the high spatial resolution of the satellite data. The resulting column-averaged dry air mole fractions are denoted by XCO.

The retrieval error sources can be grouped into systematic and random error components. Systematic errors typically occur when the analysed scenes are not well characterised by the forward model, particularly in the presence of strong scatterers under challenging conditions concerning measurement geometry and albedo. The random component is dominated by detector noise and pseudo-noise determined by specific atmospheric parameters or instrumental features. Based on a validation with ground-based Fourier Transform Spectrometer (FTS) measurements of the Total Carbon Column Observing Network (TCCON) (Wunch et al., 2011), the TROPOMI/WFMD XCO data set is characterised by a random error (precision) of $5.1\,\mathrm{ppb}$ and a systematic error (relative accuracy) of $1.9\,\mathrm{ppb}$ after quality filtering (Schneising et al., 2019).

Among others, the standard quality filter typically removes cloudy scenes and was chosen to be rather strict to meet the demanding requirements on the precision and accuracy of simultaneously retrieved $XCH_4$ globally. For example, quality-filtered ocean retrievals are mainly limited to sun glint or glitter scenes as a consequence of the otherwise weak signal above water surfaces. However, a local comparison with the cloud product from the Visible Infrared Imaging Radiometer Suite (VIIRS) instrument (Hutchison and Cracknell, 2005) onboard the joint NASA/NOAA Suomi National Polar-orbiting Partnership (Suomi-NPP) satellite for days before fire ignition has indicated that the filter can be somewhat relaxed for the present study to

maximise the number of utilisable scenes. The implemented alternative quality screening algorithm is based on simultaneously measured methane and filters scenes where the retrieved $XCH_4$ is more than 3 times the random error $\approx 50\,\mathrm{ppb}$ smaller than an assumed reference (averaged cloud-free abundances of November 5-7). The threshold was chosen to distinguish systematic from random deviations. Over weakly reflecting ocean or inland water scenes the filter is augmented by additionally flagging

scenes with large estimated CO fit error ($> 10\%$). The rationale behind the use of simultaneously measured $XCH_4$ as a quality criterion is the following. To begin with, $XCH_4$ is by far less variable than XCO in the presence of wildfires. Furthermore, both gases typically exhibit similar error characteristics regarding sign and percentage magnitude of systematic errors (Schneising et al., 2019). Hence, potential issues of the XCO data, for example due to reduced near-surface sensitivity in the presence of clouds or smoke, are clearly detected in the corresponding $XCH_4$ data and filtered out.

To get a visual impression of the smoke distribution originating from the fires, so-called true colour images (Red = Band I1, Green = Band M4, Blue = Band M3) from the VIIRS instrument are used, which show land surface, oceanic and atmospheric features like the human eye would see them (Hillger et al., 2014). The TROPOMI CO retrievals are also compared to the analysis of the ECMWF Integrated Forecasting System (IFS) provided by the Copernicus Atmosphere Monitoring Service (CAMS) (Inness et al., 2015), which assimilates MOPITT and IASI CO observations and biomass-burning emissions from the

CAMS Global Fire Assimilation System (GFAS) (Kaiser et al., 2012).

To assess the CO burden in Californian major cities we compute the average total column enhancement $E_{CO}$ (within $20\,\mathrm{km}$ radius around midtown, in units of mass per area) for the first days of the fire relative to November 7, which is considered as background. It is assumed that the additional CO from the fires is located in the well-mixed boundary layer, while the remaining upper part of the contaminated profile closely resembles the background profile, allowing to disentangle the near-

surface abundances from the total column measurements. To this end, the total column enhancement is divided by the boundary layer height $h_{bl}$ obtained from the hourly ECMWF ERA5 reanalysis product (Hersbach et al., 2018) to get the boundary layer concentration anomaly $\Delta\rho_{bl}$ due to the fires (in units of mass per volume):

$$\Delta\rho_{bl} = \frac{E_{CO}}{h_{bl}} = \frac{\Delta v_{CO} \cdot M_{CO}}{N_A \cdot A_{CO} \cdot h_{bl}} \tag{1}$$

where $\Delta v_{CO}$ is the enhancement (in units of molecules per area) relative to the pre-fire background. The molar mass of carbon

monoxide $M_{CO} = 28\,\mathrm{g\,mol^{-1}}$ and the Avogadro constant $N_A = 6.022 \cdot 10^{23}\,\mathrm{molec\,mol^{-1}}$ are used to convert molecules per area to mass per area; $A_{CO} = 0.95 \pm 0.05$ is the dimensionless near-surface CO averaging kernel characterising the boundary layer sensitivity of the retrieval determined for appropriate conditions (solar zenith angle $\in [50°, 60°]$, albedo $\in [0.1, 0.2]$). The boundary layer height determines the available volume for pollution dispersion and is thus a critical parameter for air quality assessment. The ERA5 boundary layer height is defined as the lowest height where the bulk Richardson number, which

interrelates stability with vertical wind shear, reaches the critical value of $0.25$ (ECMWF, 2018). The corresponding uncertainty estimates are based on a 10-member 4D-Var ensemble. Furthermore, an additional uncertainty associated to the estimation method of the boundary layer height is introduced, which is derived from a comparison of ERA5 boundary layer heights to lidar measurements (Wang et al., 2019). The areal variation of this anomaly is determined from the standard deviations of the CO columns measuring the inhomogeneity of the boundary layer concentrations within the respective city area.

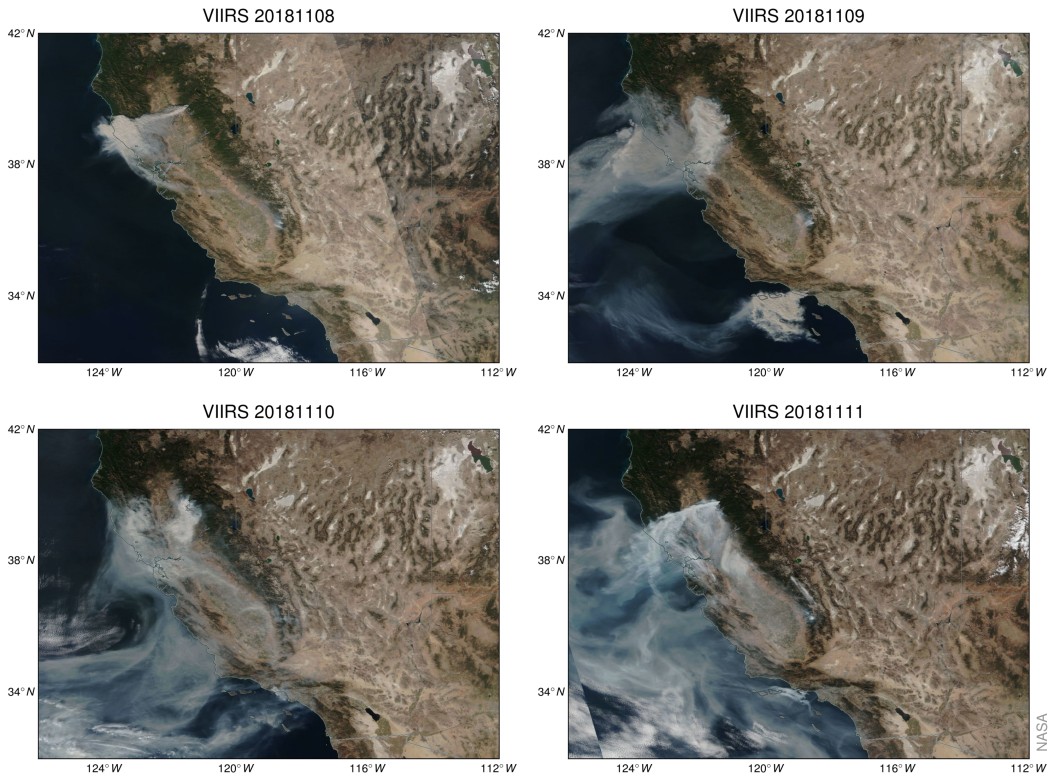

**Figure 1.** True colour reflectances from the Visible Infrared Imaging Radiometer Suite (VIIRS) for the first days of the fires taken from the NASA Worldview application.

The error analysis includes uncertainties arising from boundary layer height, the vertical distribution of emissions near the source, and smoke aerosol. Thereby, gridded Integrated Monitoring and Modelling System for wildland fires (IS4FIRES) injection heights (Sofiev et al., 2012) (corresponding to the top of the plume) as obtained from the CAMS GFAS and CAMS CO vertical profiles are used to estimate how much of the pyrogenic CO may leave the boundary layer.

## 3   Results and Discussion

### 3.1   Quality filtered XCO and comparison to CAMS

As a result of the Camp and Woolsey fires ignited on November 8, associated smoke overcast large parts of the state for nearly two weeks. This can clearly be seen on the VIIRS true colour images in Figure 1 for the first days of the fires. Sentinel-5 Precursor and Suomi-NPP fly in loose formation, with Sentinel-5P trailing behind by 3.5 minutes, ensuring that both satellites observe (almost) the same scene. Thus, the corresponding images can be compared directly.

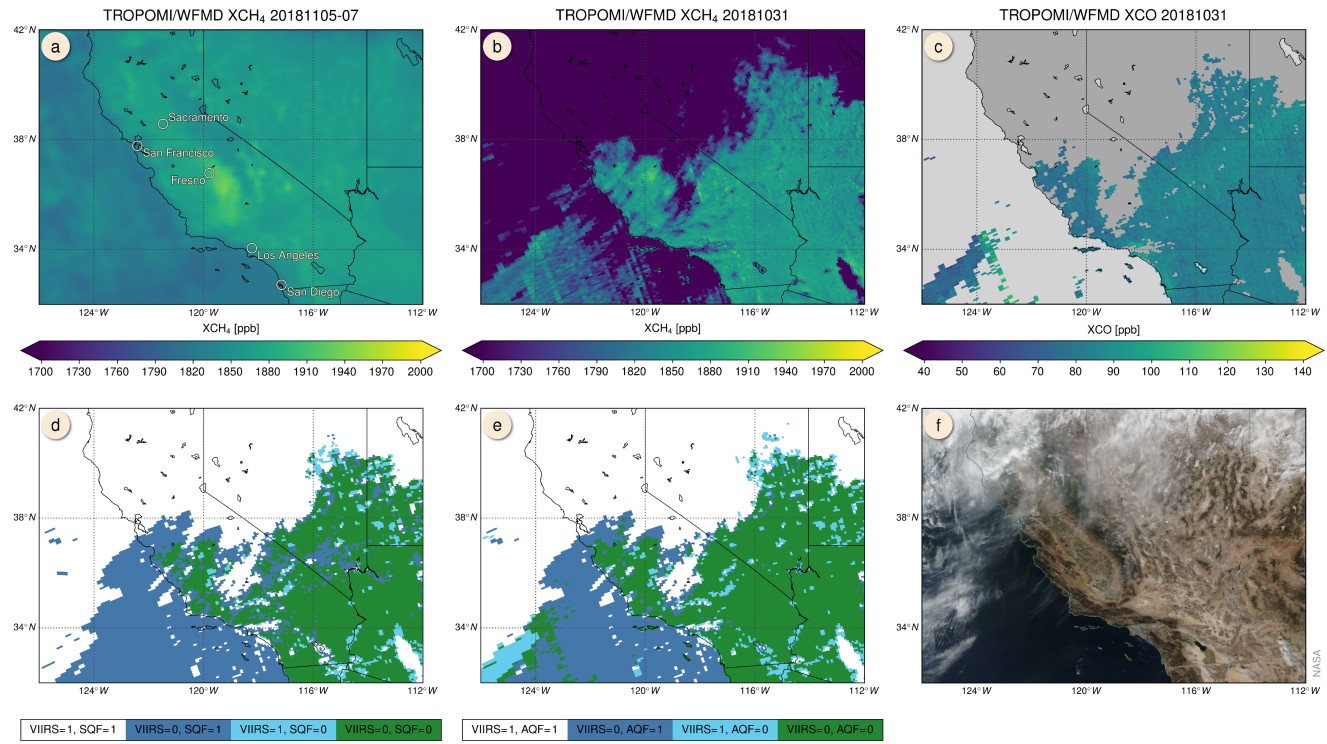

**Figure 2.** Performance of the implemented quality filter for the example day October 31. a) Cloud-free reference $XCH_4$ abundances (November 5-7). b) Unfiltered $XCH_4$ data. c) XCO after application of the filter removing scenes with unrealistic low $XCH_4$. d) Comparison of the standard quality filter (SQF, 1: excluded) with the VIIRS cloud classification (1: cloudy). Matching classifications are shown in white and green (agreement with VIIRS: 78%, passing SQF: 32%). e) As before but for the alternative quality filter (AQF) used in the presented analysis (agreement with VIIRS: 81%, passing AQF: 39%). f) VIIRS true color image.

The performance of the quality filter based on simultaneously measured methane is demonstrated in Figure 2 for an example day before the start of the analysed fires. In line with the error analysis based on synthetic data presented in Schneising et al. (2019), there typically is a considerable underestimation of $XCH_4$ in the presence of clouds due to shielding of the underlying partial columns. After application of the alternative quality filter, there are no obvious issues with the XCO data. The relaxed quality screening algorithm provides similar results to the standard filter concerning the overall agreement rate with the VIIRS cloud product, but actually yields more scenes passing the filter (about 20% for the analysed example day).

Figure 3 shows the daily $XCH_4$ distribution over California, which is used for quality screening. For each day, $XCH_4$ resembles the pre-fire background abundances shown in Figure 2a with the exception of considerable underestimations here and there mainly due to reduced near-surface sensitivity in the presence of clouds or smoke near the origin of the fires. However, in sufficient distance of the seat of fire the $XCH_4$ abundances are not affected and a quantitative analysis is still possible, even in cases where efficient scattering in the visible spectral range is indicated by extensive plumes in the VIIRS images. The explanation for this is the particle size distribution of the wildfire smoke. While clouds typically consist of water droplets

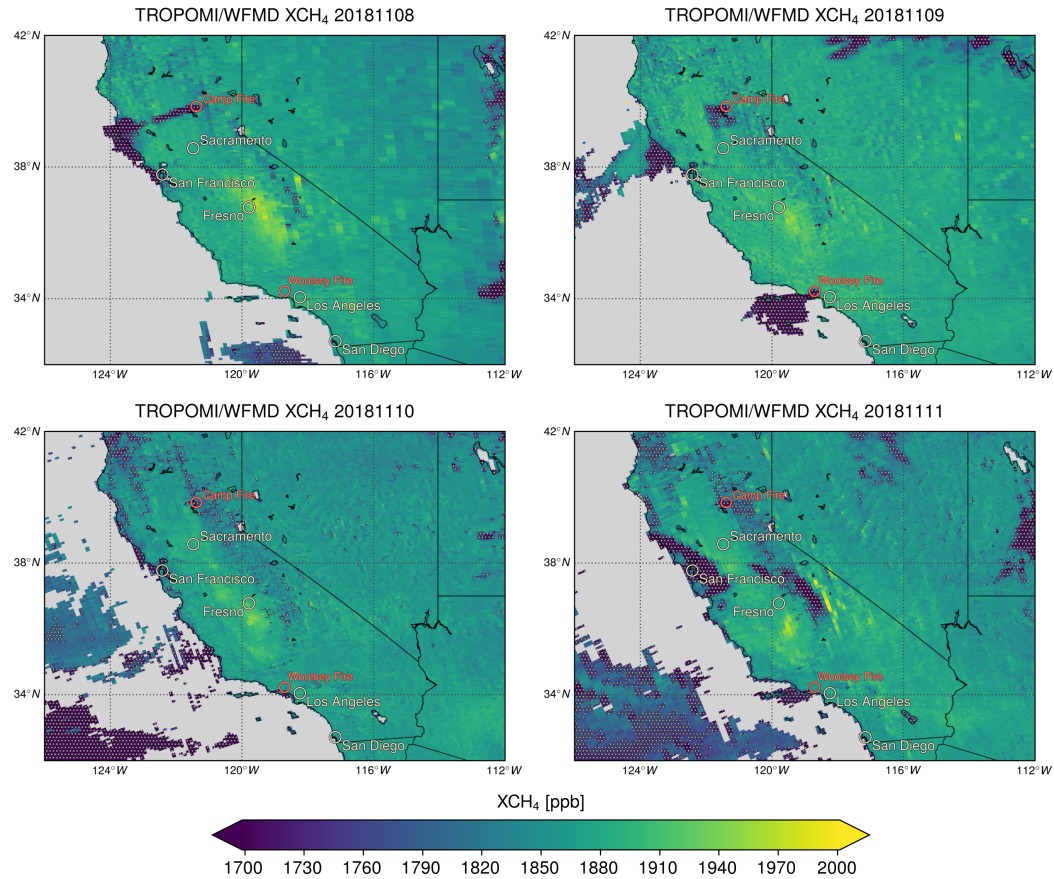

**Figure 3.** Retrieved methane column-averaged mole fractions from TROPOMI for the same days as in Figure 1. The $XCH_4$ is used to filter out scenes with significant underestimation (dotted scenes) mainly due to reduced near-surface sensitivity in the presence of clouds or smoke due to shielding of the subjacent partial columns. The Central Valley exhibits combined anthropogenic methane emissions from oil fields and agriculture (Schneising et al., 2019).

with an effective radius of the order of $10\,\mu m$, smoke is dominated by considerably smaller particles. The mass distribution of smoke plumes shows a prominent peak at about $0.3\,\mu m$ (Stith et al., 1981) but is nevertheless dominated by a small number of supermicron-sized particles including some very large particles (Radke et al., 1990). As a consequence of the different size distributions, clouds have a typical Ångström exponent $\alpha = 0$ and thus no wavelength dependence of the aerosol optical depth, while biomass burning aerosols have a distinct wavelength dependence with typical $\alpha$ ranging between 1 and 2 depending on the fire (Eck et al., 2009). The submicron particles reduce the visibility and lead to extended smoke plumes over large distances in the VIIRS true color reflectances shown in Figure 1. However, the $2.3\,\mu m$ spectral range, where the satellite measurements are taken, is subject to only little scattering at these small particles. The satellite retrievals close to the source of the fire are rather affected by the large supermicron-sized particles, which have a short atmospheric lifetime tending to fall out rapidly

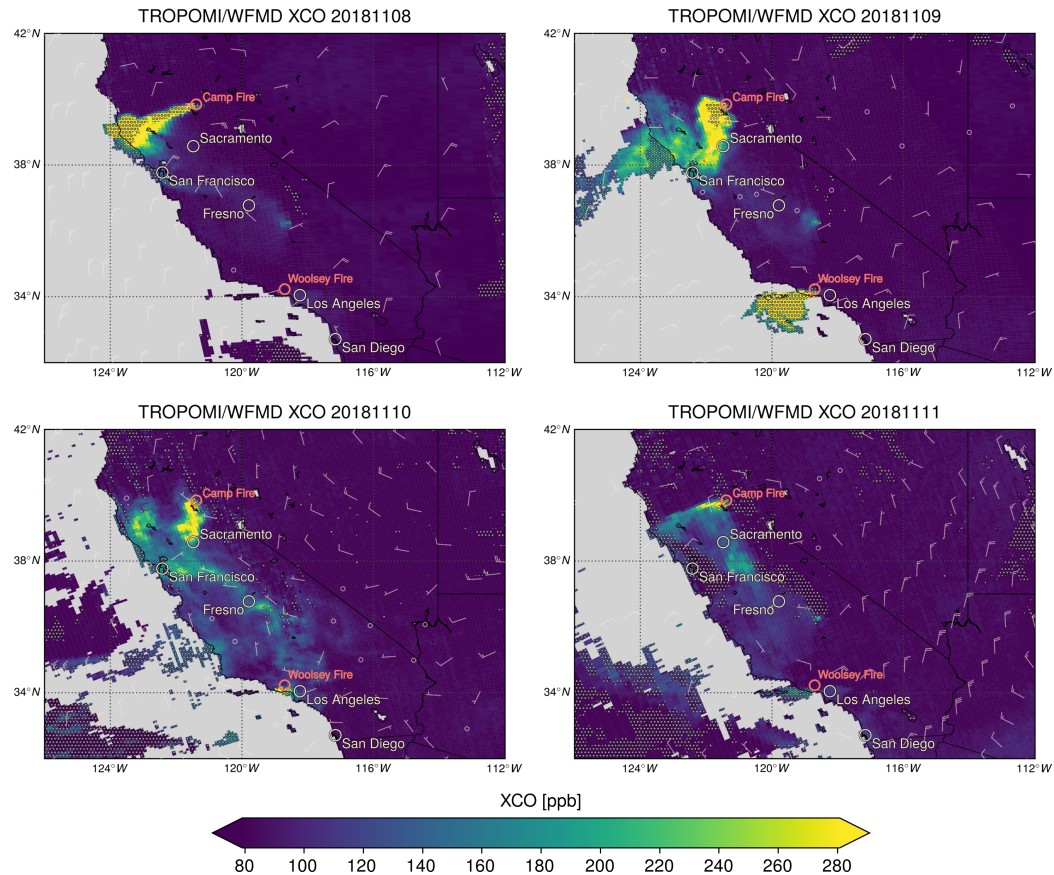

**Figure 4.** Retrieved carbon monoxide column-averaged mole fractions from TROPOMI for the same days as in Figure 1. Dotted scenes are excluded by the quality filter based on simultaneously retrieved XCH₄. Also shown is the mean wind in the boundary layer obtained from ECMWF data.

(World Health Organization, 2006) and are thus getting more and more negligible when departing from the seat of fire. Thus, a reliable XCO retrieval is possible in smoke plumes in the far field of the fire origin for scenes passing the quality filter. Corresponding simulations with realistic aerosol optical depth and Ångström exponent are included in the error analysis in the next subsection to quantify the impact of scattering at smoke aerosols.

5    Figure 4 shows the XCO distribution over California, which matches the smoke emission and transport patterns detected by VIIRS unambiguously. This substantiates that the observed CO enhancements are actually originating from the wildfires. It can be seen that the abundances over the major cities we want to analyse are typically not filtered out and are thus suitable for a quantitative analysis. However, the quantitative interpretation of scenes right above or too close to the origin of the fire is limited by reduced vertical retrieval sensitivity near the surface and are consequently filtered out.

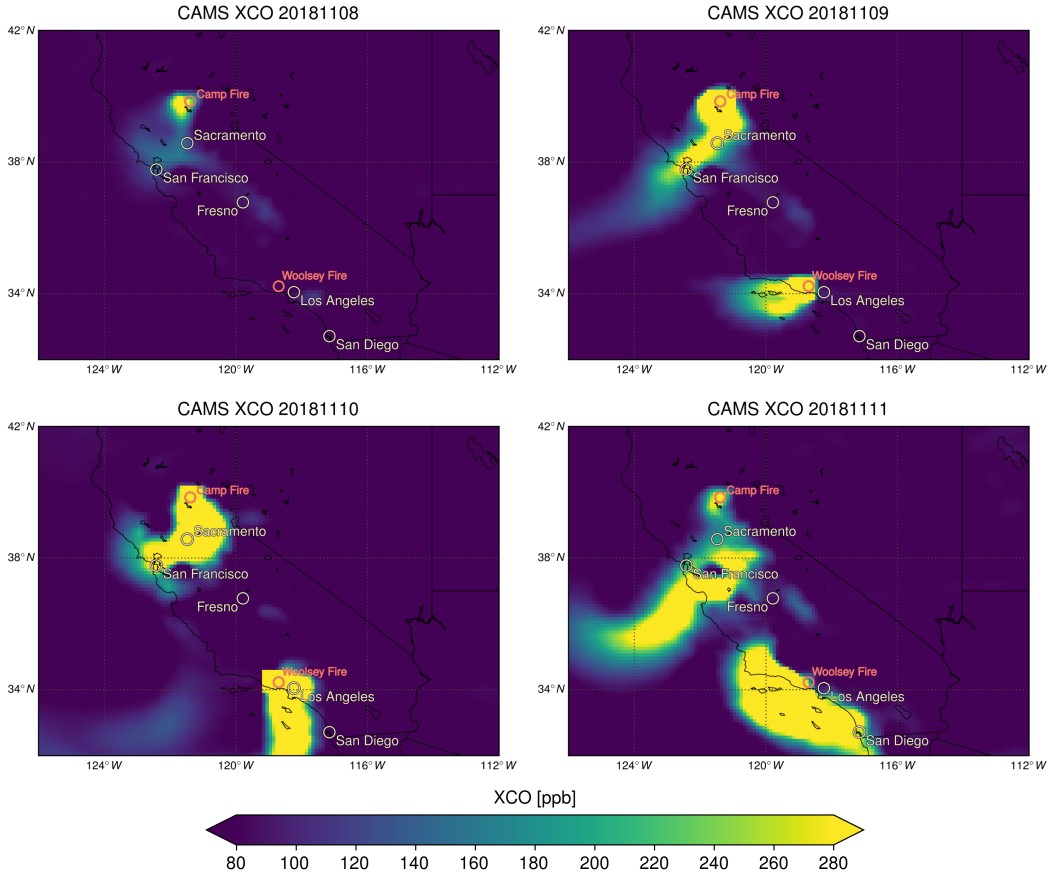

**Figure 5.** CAMS near-real time CO analysis for the first days of the fires at 24:00 UTC corresponding to 16:00 local time (Pacific Standard Time).

For comparison, Figure 5 shows the CAMS near-real-time CO analysis on a $0.1° \times 0.1°$ grid for the same days shown in the previous figures and the closest available time to the TROPOMI overpass at 13:30 local time. As CAMS is available in time steps of 6 hours, the analysis corresponding to 16:00 local time is used for the comparison. Although CO emissions from the fires are obviously included in the CAMS data, the transport patterns seem to be somewhat different. While the patterns are
5    broadly consistent for November 9 and 10, the modelled wind fields close to the fires seems to deviate on November 8 and 11, which results in a longer continuance of the plume over land, while the VIIRS images and the TROPOMI data suggest a faster transport westwards to the sea. This can also be seen in Figure 6 showing departures of the CAMS analysis from the TROPOMI XCO after averaging the satellite data on the CAMS resolution. Apart from the partially different transport patterns, also the intensity distribution close to the fire sources is different, with CAMS abundances being considerably higher for the
10   most part and deviations reaching several hundred ppb. This may be due to overestimated wildfire fluxes, underestimated initial horizontal transport in the vicinity of the fire sources, or a combination of both.

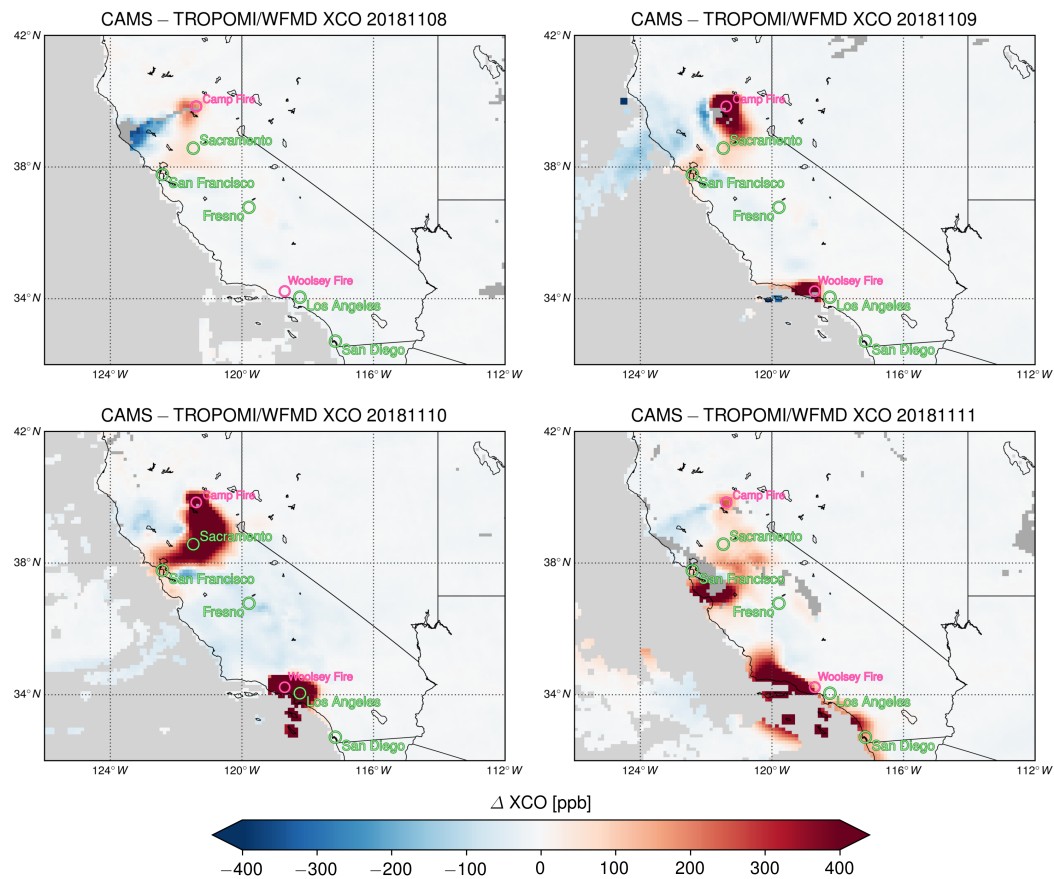

**Figure 6.** Difference of CAMS XCO analysis to TROPOMI/WFMD satellite measurements.

## 3.2 Boundary layer concentration anomalies and associated uncertainties

To assess the CO burden in Californian cities, the boundary layer CO concentration anomaly is computed according to Equation 1. The diurnal variation of the ECMWF ERA5 boundary layer heights and their inherent uncertainties are illustrated in Figure 7. There is a strong diurnal cycle with low values at night and maximal values around local noon close to the time

5    of the TROPOMI overpass at 13:30. The boundary layer concentration uncertainty arising from boundary layer height $\sigma(h_{bl})$ is determined from the maximal ERA5 ensemble uncertainty between 13:00 and 14:00 local time and the variation within this hour in each case. Typical values of $\sigma(h_{bl})$ range between $10\%$ and $25\%$. The ERA5 boundary layer height around the satellite overpass time is about $300\,\mathrm{m}$ smaller than the boundary layer height derived from aerosol and turbulence detection lidar measurements by different retrieval methods (Wang et al., 2019). This uncertainty associated to the estimation method is

10    additionally taken into account in the error budget by an extra term $\sigma_m(h_{bl})$ quantifying the percentage impact on the smaller end of $\Delta\rho_{bl}$.

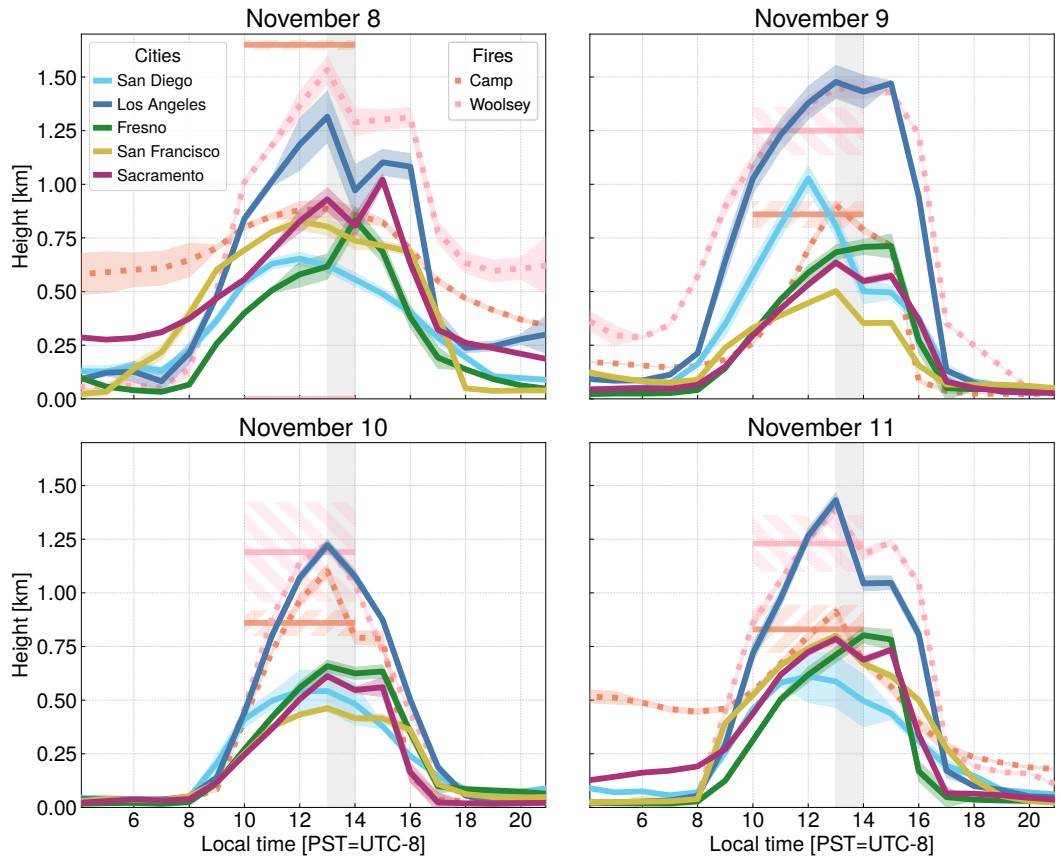

**Figure 7.** Diurnal variations of the boundary layer heights obtained from the ECMWF ERA5 reanalysis for major Californian cities (solid) and fires (dotted). The uncertainty estimates are based on a 10-member 4D-Var ensemble; the additional uncertainty associated to the estimation method of the boundary layer height is not shown here. Also shown are mean IS4FIRES smoke injection heights and their variation for both analysed fires as horizontal bars and surrounded hatched areas. The grey-shaded area illustrates the TROPOMI overpass time. On November 8 the injection height of the Woolsey Fire is zero because it started later in the day.

The potentially largest source of uncertainty in calculation of the boundary layer CO burden is plume dynamics and the question if all CO remains in the boundary layer or if a certain proportion reaches the free troposphere. The vertical distribution of emissions near the source is driven by the fire radiative power and the local ambient atmospheric conditions such as stability and humidity. Three types of wildfire plumes are distinguished by the amount of condensed water vapour during

plume formation (Fromm et al., 2010): 1) Dry smoke plumes, which contain water vapour and usually stay within the boundary layer, 2) Pyrocumulus containing water droplets either staying in the boundary layer or reaching the free troposphere depending on atmospheric conditions, and extreme 3) Pyrocumulonimbus scenarios containing ice particles and potentially reaching the stratosphere. Typically, most of the biomass burning emissions stay within the mixing layer and cases with pyro-convection or direct injection to the free troposphere or even higher are rare (Labonne et al., 2007; Mazzoni et al., 2007; Tosca et al., 2011).

As can be seen in Figure 7, the IS4FIRES injection heights corresponding to the top of the plume are equal or smaller than the respective maximum boundary layer height at the location of the fires (all the more when considering the additional uncertainty associated to the estimation method of the boundary layer height not shown in the figure), with the exception of the first day of the Camp Fire. This sole discrepancy may be linked to overestimated fire radiative power for the Camp Fire on the day of ignition, which is also suggested by the comparison of the CAMS XCO analysis to the TROPOMI retrievals

showing considerably higher abundances for CAMS in the vicinity of the fire source. In summary, the IS4FIRES injection height analysis indicates that most of the CO load stays within the boundary layer. Furthermore, the entire state of California was at least abnormally dry within the analysed time period with a moderate drought at the Camp Fire origin and a severe drought at the seat of the Woolsey Fire according to the United States Drought Monitor (https://droughtmonitor.unl.edu/). These are favourable conditions for dry smoke plumes being trapped in the boundary layer also rendering later deep moist

convection with transport to the free troposphere during the first days of the fire at another location unlikely. Finally, there is also no indication for Pyrocumulus or Pyrocumulonimbus in the VIIRS true color images as there is no obvious cloud formation over the fires (see Figure 1).

Nevertheless, partial venting to the free troposhere cannot be entirely excluded and we therefore introduce an uncertainty $\sigma(h_{inj})$ arising from unknown plume dynamics, which is only applied to the smaller end of the boundary layer concentration

anomaly because lost CO of this type can only lead to an overestimation of the near-surface concentrations. $\sigma(h_{inj})$ for the analysed cities is estimated by the CO mass fraction $f_a$ above the upper bound of the ERA5 boundary layer height uncertainty range as determined from the CAMS model CO vertical profiles (see Figure 8). As there is no indication that the CO fraction in the free troposphere grows significantly during the analysed period over the cities considered, mean CAMS profiles for days with substantial CO enhancement are examined. The CAMS CO profile analysis further reinforces the assumption that most

of the emitted CO stays within the boundary layer even four days after ignition and at a greater distance from the fire sources. $f_a$ is below $10\%$ with the exception of San Diego. Thus, the uncertainty $\sigma(h_{inj})$ is set to $30\%$ for San Diego and to $10\%$ for the other cities.

Another potential error source associated with fires is smoke aerosol. Scenes with reduced near-surface sensitivity due to clouds and smoke with large particles near the seat of the fires are automatically filtered out using simultaneously measured

methane. Figure 3 also demonstrates that methane is not considerably increased compared to the pre-fire background abun-

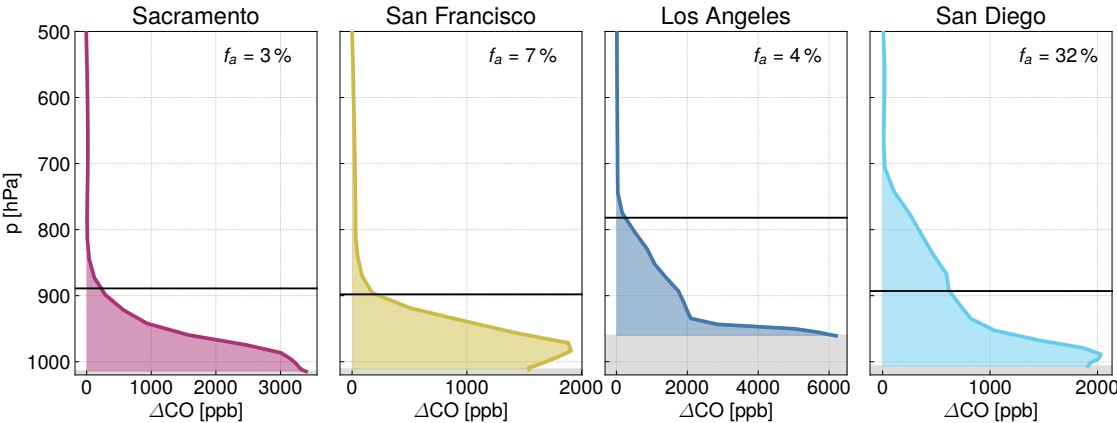

**Figure 8.** Mean CAMS CO profile enhancement relative to the pre-fire background (November 7). Fresno is not shown because there is no significant enhancement during November 8-11 in the CAMS model. The upper edges of the grey areas represent the surface pressure, the black vertical lines illustrate the upper bound of the uncertainty range of the estimated mean ERA5 boundary layer height, and $f_a$ is the CO mass fraction above the boundary layer.

dances (Figure 2a) and that the XCO enhancement patterns are not resembled in $XCH_4$. Thus, it can be excluded that the detected XCO enhancement is only an artefact as a result of light path lengthening because of aerosol scattering at the particulate matter of the smoke, because such systematic errors would affect both retrieved gases similarly.

To assess the potential impact of smoke aerosol quantitatively, simulated measurements are used. This means that sun-normalised radiances for an assumed smoke scenario are calculated with the radiative transfer model, which are subsequently used as measurement input in the retrieval. The errors are then defined as the deviation of the retrieved columns for the smoke scenario from the corresponding columns for the background scenario also used to calculate the forward model look-up table. To model wildfire conditions in sufficient distance from the seat of the fire with low visibility but decreasing scattering issues at larger wavelengths (consistent with Figures 1 and 3) we use the *extreme in BL* aerosol scenario originally introduced in Schneising et al. (2008) containing urban aerosol with a significant soot fraction (Shettle and Fenn, 1979) combined with an extreme CO profile with an 10-fold enhancement in the boundary layer compared to the standard profile. The used aerosol scenario (aerosol optical depth $\tau_{550\,nm} \approx 3$ and Ångström exponent $\alpha \approx 1$) is considered a realistic worst case scenario for the analysed fires because it is at the upper end of optical depths and at the lower end of Ångström exponents for typical fire aerosols (Eck et al., 2009). Furthermore, the corresponding aerosol profile is consistent with the previous results about the vertical distribution of the emitted species during the first four days of the fires. Thus, the Camp Fire and the Woolsey Fire very likely exhibit less scattering in the $2.3\,\mu m$ spectral range than our model scenario assumes at least during the period analysed. The corresponding results are summarised in Figure 9. Typical systematic CO errors for Californian cities on the analysed days range between about $-3\%$ and $2\%$ for the assumed aerosol type and CO profile. Therefore, the uncertainty due to smoke aerosol $\sigma(a_{smo})$ is set to $5\%$ adding an extra amount due to the uncertainty of the actual aerosol type.

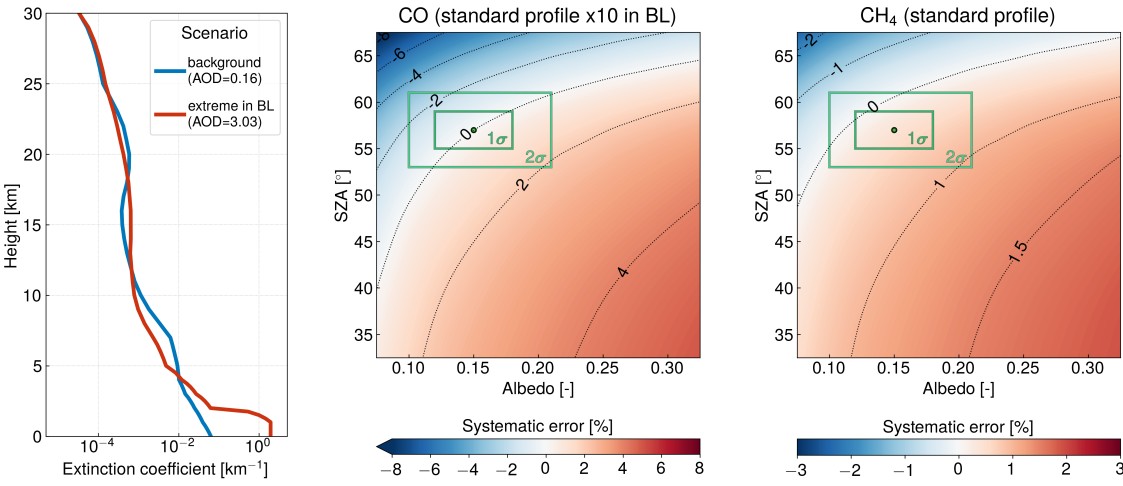

**Figure 9.** The left panel shows the aerosol extinction profiles used in the analysis of smoke aerosol errors. Also given are the corresponding aerosol optical depths at $550\,\text{nm}$. The other panels show the systematic errors of CO and $CH_4$ as function of solar zenith angle and albedo when using the extreme instead of the background aerosol scenario. The green boxes highlight the typical conditions for the Californian cities on the analysed days using percentiles corresponding to $1\sigma$ (68% of data) and $2\sigma$ (95% of data). The green circle is the pair of median albedo and median solar zenith angle.

The total uncertainty of the boundary layer concentration anomaly $\sigma(\Delta\rho_{bl})$ is determined by

$$\sigma^2(\Delta\rho_{bl}) = \sigma^2(A_{\text{CO}}) + \sigma^2(h_{bl}) + \sigma_m^2(h_{bl}) + \sigma^2(h_{inj}) + \sigma^2(a_{smo}) \qquad (2)$$

Averaged boundary layer concentration anomalies of CO (relative to November 7) in major Californian cities during the first days of the Camp and Woolsey fires are presented in Figure 10 together with the total uncertainty of Equation 2 and an estimate of the areal variation measuring the inhomogeneity of the CO concentrations within the city area. The largest values are found for Sacramento and San Francisco on November 9 and 10 due to the prevailing wind conditions with boundary layer concentration anomalies of about $2.5\,\text{mgCO}\,\text{m}^{-3}$, which is well below the national CO air quality standard of $10\,\text{mg}\,\text{m}^{-3}$ even after adding a typical background concentration of about $0.5$–$1.0\,\text{mgCO}\,\text{m}^{-3}$. The cities in the southern part of the state are less affected owed to more favourable weather conditions.

Although the Sacramento and San Francisco city averages are compliant with air quality standards, the large associated areal variations indicate an uneven CO distribution within both towns, in particular for Sacramento. This interpretation is supported by the CO distribution depicted in Figure 4 showing that the plume's edge of the Camp Fire is located near Sacramento leading to a larger burden in the northwest compared to the rest of the city.

The largest burden with respect to CO within all city radii is actually found on November 10 about $10\,\text{km}$ to the east of Sacramento International Airport, where one finds a considerable column enhancement of $3.14\,\text{g}\,\text{m}^{-2}$. Given the ECMWF ERA5 boundary layer height of $580\,\text{m}$, this corresponds to a boundary layer concentration anomaly of $5.42\,\text{mgCO}\,\text{m}^{-3}$ [$3.41$–$6.00; 1\sigma$]. The largest enhancement on November 9 is also located in the vicinity of Sacramento Airport (about $10\,\text{km}$ to

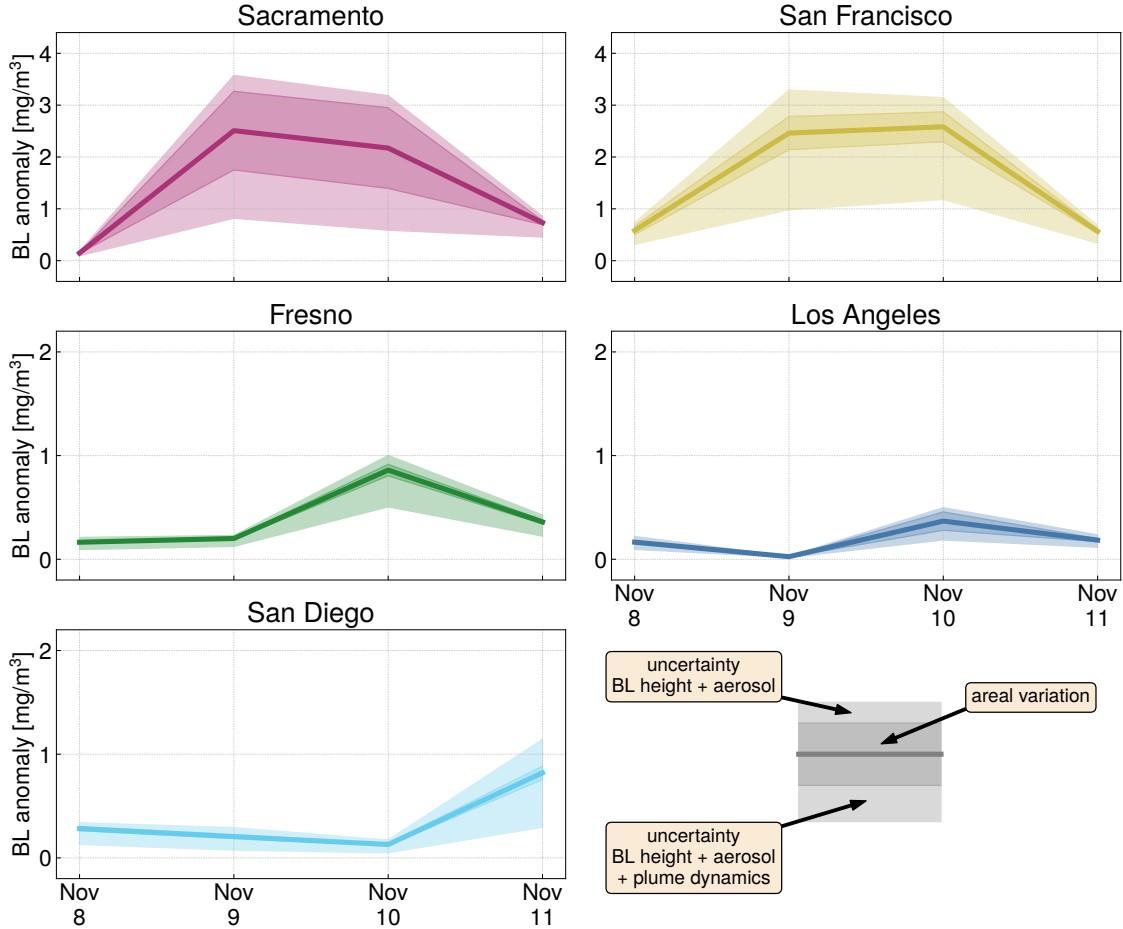

**Figure 10.** Averaged boundary layer concentration anomalies of CO (relative to November 7) and associated areal variations and uncertainties ($1\sigma$) in major Californian cities during the first days of the Camp and Woolsey fires.

the southwest) and amounts to $3.13\,\mathrm{g\,m^{-2}}$ with a boundary layer height of $592\,\mathrm{m}$ leading to a boundary layer concentration anomaly of $5.28\,\mathrm{mgCO\,m^{-3}}$ [3.32–5.93; $1\sigma$]. Thus, the national ambient air quality standard of $10\,\mathrm{mgCO\,m^{-3}}$ was likely not exceeded even for the most polluted city scenes and after adding a typical background of about $0.5$–$1.0\,\mathrm{mgCO\,m^{-3}}$.

To further assess the described area with significantly increased boundary layer concentrations, we revisit the discussed contaminated scene near Sacramento International Airport on November 10 and analyse associated results from CAMS and ground-based Quality Assurance Air Monitoring Site Information. For the grid-box comprising the mentioned satellite scene, CAMS predicts a considerably larger column enhancement of $5.93\,\mathrm{g\,m^{-2}}$ corresponding to a boundary layer concentration anomaly of $10.23\,\mathrm{mgCO\,m^{-3}}$ [6.51–11.04; $1\sigma$] using the ERA5 boundary layer height and the associated uncertainties. Although this is almost twice as high as the satellite derived concentration anomaly and potentially exceeds the national ambient

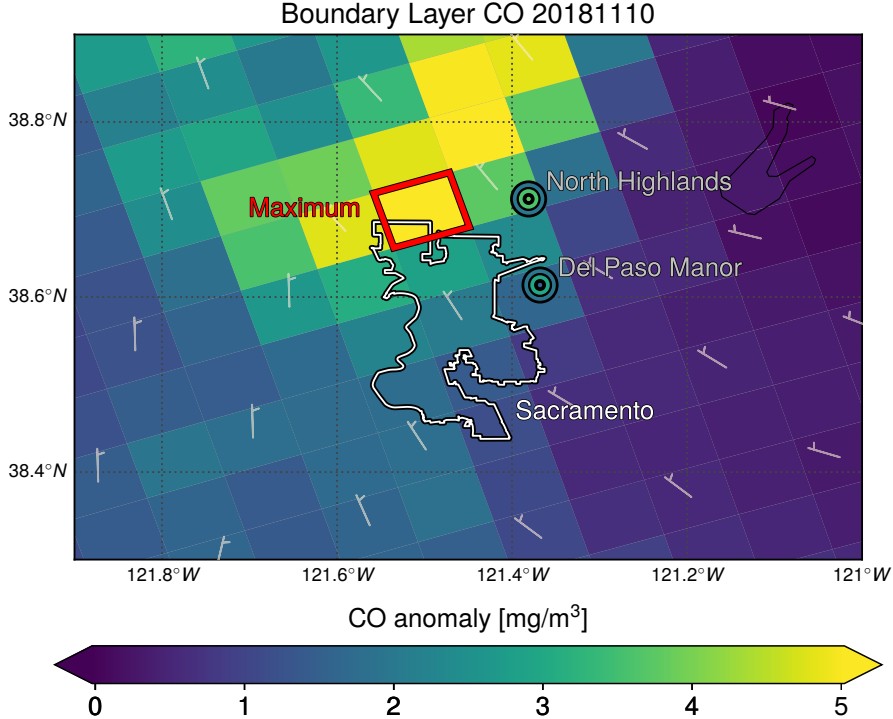

**Figure 11.** Boundary layer concentration anomalies of Sacramento and its environs determined from TROPOMI CO total column measurements and boundary layer heights from the ECMWF ERA5 reanalysis. Highlighted are the satellite scene with maximal city area value ($5.42\,\mathrm{mgCO\,m^{-3}}$ [3.41–6.00; $1\sigma$], red) and the location of the AQMIS sites in North Highlands and at Del Paso Manor (black). The anomalies based on the maximum values of the ground-based sites (3.60 and $3.00\,\mathrm{mgCO\,m^{-3}}$) are colour-coded in the inner circle at the site location, the anomalies based on the daily averages ($1.93\,\mathrm{mgCO\,m^{-3}}$ each) are colour-coded in the outer circle.

air quality standards, the error bars of the CAMS and satellite derived concentration anomalies almost overlap due to the relatively large uncertainties arising from boundary layer height estimation and unknown plume dynamics.

Ground-based measurements are available from the Air Quality and Meteorological Information System (AQMIS) network (California Air Resources Board, 2018) providing daily maximum and daily average values. There are three CO measurement sites in Sacramento County. For the site at Bercut Drive in Sacramento the data set is incomplete during the first days of the fire and therefore excluded from the comparison. The second site at Blackfoot Way in North Highlands is located farther north and closer to the analysed contaminated satellite scene. The maximum value during the first four days of the fire is stated to be $4.1\,\mathrm{ppm}$ ($4.8\,\mathrm{mgCO\,m^{-3}}$) on November 10. Relative to the maximum value of November 7 this corresponds to a concentration anomaly of $3.60\,\mathrm{mgCO\,m^{-3}}$. The third site is at Del Paso Manor in Sacramento with an maximum value of $3.8\,\mathrm{ppm}$ ($4.4\,\mathrm{mgCO\,m^{-3}}$) on November 10 corresponding to a concentration anomaly of $3.00\,\mathrm{mgCO\,m^{-3}}$. When using the daily averages instead of the maximum values, the concentration anomalies amount to $1.93\,\mathrm{mgCO\,m^{-3}}$ for both sites.

Figure 11 shows the boundary layer concentration anomalies of Sacramento and its surrounding districts allowing to get an overview of the situation by highlighting the locations of the different measurements. As can be seen, the AQMIS sites are located easterly of the satellite scene with maximal city area CO value, where concentrations are beginning to decline steeply. The corresponding satellite averages at both analysed AQMIS sites are broadly consistent with the ground-based measurements taking into account the potential variability within a satellite scene indicated by the scene-to-scene gradient of the satellite data and the fact that the sites are located at the edge of satellite scenes. While the ground-based anomaly based on the maximum values in North Highlands matches well with the value of the associated satellite scene, the ground-based anomaly based on the daily averages rather resembles the values of adjacent satellite scenes to the east or to the south. At Del Paso Manor the opposite is true: the ground-based anomaly based on the daily averages fits the surrounding satellite scene well, while the anomaly based on the maximum values rather matches the adjacent satellite scene to the north.

## 4   Conclusions

We have performed an analysis of atmospheric carbon monoxide (CO) concentration changes introduced by emissions of fires using measurements in the shortwave infrared spectral range of the TROPOMI instrument onboard the Sentinel-5 Precursor satellite. The local CO emissions of Californian wildfires and subsequent transport can be clearly observed from space. Due to its unique features, CO retrievals from TROPOMI have the potential to trigger model improvement and a better quantification of fire emissions by assimilation of the satellite-derived XCO in integrated systems such as CAMS.

Furthermore, new fields of application are enabled, in particular the detection of emission hot spots or air quality monitoring tasks, because large sources are readily detected in a single overpass. The evaluation of TROPOMI's capabilities for dense air quality monitoring has shown that the quantitative assessment of the CO burden in Californian major cities is possible on a daily recurrent basis using the example of the first days of the Camp Fire and Woolsey Fire in November 2018.

However, the accurate determination of boundary layer concentrations depends on reliable external mixing layer height information. In the case of fires, the feasibility is also subject to specific favourable circumstances affecting the vertical distribution of emissions. The local ambient atmospheric conditions such as stability and humidity have to ensure that most of the fire emissions stay within the boundary layer and that pyro-convection or direct injection to the free troposphere is unlikely. As a consequence, unknown plume dynamics generally remains the largest source of uncertainty in the calculation of the boundary layer CO burden caused by wildfires. The case study presented here benefits from the fact that most of the emitted CO remains in the boundary layer during the analysed period, which considerably reduces the associated uncertainties.

The quantitative analysis has shown that even intense wildfire events are not necessarily associated with the exceedance of national ambient air quality standards in the far field of the fires because all major city scenes for the analysed days comply with the regulatory limits. This finding is also confirmed by isolated ground-based air quality measurements near the most polluted city scenes.

Increasing unusual weather conditions with dryness of vegetation on the rise may lead to longer lasting and more intense fire seasons in the future. Therefore, it is getting more and more important to monitor and forecast the air quality decline

associated with wildfires in a changing climate to evaluate whether the compliance with regulatory limits will last or not. This can be achieved by an integrated monitoring system combining modelling with complementary information from accurate ground-based measurements and observations from various satellites.

*Data availability.* The carbon monoxide and methane data sets presented in this publication can be accessed via http://www.iup.uni-bremen.
de/carbon_ghg/products/tropomi_wfmd/.

*Author contributions.* OS: writing the paper, design and operation of the satellite retrievals, data analysis, interpretation. MB, MR, HB, JPB: significant conceptual input to writing, design of the satellite retrievals, interpretation. All authors discussed the results and commented on the manuscript.

*Competing interests.* The authors declare that they have no conflict of interest.

*Acknowledgements.* This publication contains modified Copernicus Sentinel data and modified Copernicus Climate Change Service Information (2018). Sentinel-5 Precursor is an ESA mission implemented on behalf of the European Commission. The TROPOMI payload is a joint development by ESA and the Netherlands Space Office (NSO). The Sentinel-5 Precursor ground-segment development has been funded by ESA and with national contributions from The Netherlands, Germany, and Belgium. The research leading to the presented results has in part been funded by the ESA projects GHG-CCI, GHG-CCI+, and S5L2PP, the Federal Ministry of Education and Research project
AIRSPACE, and by the State and the University of Bremen.

We acknowledge the use of VIIRS imagery from the NASA Worldview application (https://worldview.earthdata.nasa.gov/) operated by the NASA/Goddard Space Flight Center Earth Science Data and Information System (ESDIS) project, the use of data from the California Department of Forestry and Fire Protection and the U.S. Environmental Protection Agency. We also thank the European Centre for Medium-Range Weather Forecasts (ECMWF) for providing the meteorological analysis, the ERA5 reanalysis, as well as the Copernicus Atmosphere
Monitoring Service (CAMS) carbon monoxide analysis and Global Fire Assimilation System (GFAS) data. Neither the European Commission nor ECMWF is responsible for any use that may be made of the Copernicus Information or Data it contains. Ground-based CO data was obtained using the Air Quality Data (PST) Query Tool (https://www.arb.ca.gov/aqmis2/aqmis2.php) of the Air Quality and Meteorological Information System (AQMIS).

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
