# Peer review of "Severe Californian wildfires in November 2018 observed from space: the carbon monoxide perspective"

_Atmospheric Chemistry and Physics, 2019_

## Referee Comment (RC1) · Anonymous Referee #1 · 22 Feb 2019

The paper by Schneising et al. reports on the detection of the carbon monoxide (CO) plumes from the 2018 Californian wildfires by the new TROPOMI satellite instrument. The study attempts to derive conclusions for the air quality burden by translating column enhancements into concentrations under the assumption that boundary layer height is known and that all measured CO enhancements reside in the boundary layer.

General comments

1. The style of the text is at the edge of what is acceptable for scientific writing.

It uses emotional and judgemental wording, (non-exhaustive) list of examples: title: "devastating"; abstract: "one of the most disastrous months in Californian history",

"destructive wildfires raging", "burnt to cinders"; introduction: "the town of Paradise was wiped out", "an unprecedented instance in history"; conclusion: "The analysed fires were the latest episodes of the deadliest and most destructive wildfire season the state of California has ever faced." Most of these statements can be removed without loss of any information.

Further, the manuscript is very short in making reference to previous work. More references to earlier CO work of the MOPPIT, SCIAMACHY, IASI, TES, AIRS teams are required.

2. What is the scientific value of the paper? The general CO detection capabilities of TROPOMI have been published before [e.g. Borsdorff et al., 2018a,b]. Air quality issues with wild-fire CO emissions are well-known. I would argue that the scientific value is the quantitative estimation of the CO burden (in units mg m-3) based on daily recurrent satellite data i.e. the evaluation of TROPOMI's capabilities for dense CO-related air quality monitoring. Comparison to the CAMS model could also be an added value since it might trigger model improvement. Currently, the methodological evaluation and model comparisons are too short and too vague to serve any of these scientific purposes.

Comments

P4, L11: Please add a discussion on errors coming from the assumptions on boundary layer height knowledge. Discuss how boundary layer height is determined. Please also add figures or tables for typical boundary layer heights. Are boundary layer heights of a few hundred meters (at midday) realistic (P7, L5)? These boundary layer heights need to be validated. If the boundary layer is so shallow, a large fraction of the fire emissions might reach above the boundary layer due to initial thermal rise.

P 4, L27: Please add and discuss a figure showing "the fact that the simultaneously retrieved gases, methane and water vapour, are not considerably increased compared to the pre-fire background abundances."

P4, L30: The CAMS comparison is too short to be of scientific value. Please add a quantitative discussion (e.g. average TROPOMI on CAMS resolution and calculate departures).

P9, L24: None of the statements in the second paragraph of the conclusion are actually conclusions based on the scientific results of the paper, but rather they are author interpretation of climate change impacts.

Figure 1: This figure could be dropped. The total column sensitivity of the solar absorption concept is standard scientific knowledge. Does the algorithm take into account that the near-ground sensitivity might be reduced due to scattering layers such as wildfire particulate plumes (or low clouds)? If not, what is the impact on the air quality derivations – does the satellite "see" the entire column?

Figures 6 and 7: The figures are too dense with internal information. None of the acronyms (VC, SZA, VZA, dlnI/dx. . .) and few of the terms (sun-normalized radiance) are explained, the panels are too small, some of the panels are not even discussed (Temperature fit, . . .). Recommendation: Either remove the figures entirely or just show the relevant parts e.g. the CO panels.
* * *

---

## Referee Comment (RC2) · Anonymous Referee #2 · 26 Feb 2019

Reviewer's Comments on "Devastating Californian wildfires in November 2018 observed from space: the carbon monoxide perspective" by Schneising et al.

General Comments

This manuscript by Schneising et al. presents some interesting observations of satellite-observed CO over California during a recent episode of extensive wildfires along with a brief analysis. However, due to several major limitations of this work, the presented results are truly only useful for qualitative analysis and are therefore of limited scientific interest. For this reason, I can not recommend publication in ACP.

The first issue is the lack of any discussion of the retrieval algorithm or the expected

error characteristics. References are provided to two ESA technical reports, but these appear not to be peer-reviewed, nor publicly available. A detailed presentation of the retrieval algorithm (in the peer-reviewed literature) is essential for establishing the provenance of the TROPOMI-WFMD CO products. While the general aspects of the algorithm might be similar to published algorithms developed for SCIAMACHY, some details will certainly be instrument-specific. Such details are extremely important. Similarly, details of the filtering methods (based on CO fit and water vapor absorption) will also influence the scientific interpretation of the data and should therefore also be discussed fully. (For example, how were the thresholds for CO fit and water vapor absorption determined?)

The second major area of concern is the lack of any proper validation results; the only mention of validation is a reference to an unpublished technical report. The history of satellite remote sensing demonstrates clearly that satellite products can not simply be taken 'at face value'. For satellite CO products, validation should preferably exploit in-situ CO vertical profiles measured from aircraft. If that approach is not feasible for some reason, the authors could either exploit ground-based FTIR CO retrievals or other satellite CO products. (These latter methods are less optimal than in situ-based methods because of issues related to averaging kernels.) Comparisons of satellite CO products with surface measurements of CO concentration are generally inadequate for validation because of the variability of CO in the middle and upper troposphere.

Beyond these two major issues, it is not clear how the TROPOMI-WFMD CO product relates to the TROPOMI-SICOR product (as developed by Borsdorff et al.). Do the two algorithms give the same results? Are there other issues which might make one product preferable over the other? Is the TROPOMI-WFMD CO product routinely generated and publicly available? These are inevitable questions that will concern potential users.

My advice to the authors is to strongly consider writing and submitting a validation paper (to AMT or another appropriate journal) which directly addresses these issues. Such a paper is an essential prerequisite to the quantitative use of satellite CO data.

Publication of that paper would pave the way for this paper and increase its significance.

Specific Comments

In several places, word choice could be improved to be less sensational and more scientific. For example, in the title, 'Devastating' could be 'Severe'. Similarly, the expression 'burnt to cinders' is gratuitous.

p. 2, line 8. Need reference for physiological effects of CO on humans.

p. 2, line 15. The text in this paragraph suggests that MOPITT and IASI CO retrievals are generally insensitive to CO near the surface, but this is overly simplistic. In fact, publications document that both of these instruments can provide useful sensitivity to CO near the surface in daytime scenes over land (i.e., in conditions of high thermal contrast). For example, see "Sensitivity of MOPITT observations to carbon monoxide in the lower troposphere," JGR, 112, doi:10.1029/2007JD008929 (2007) by Deeter et al., and "IASI's sensitivity to near-surface carbon monoxide (CO): Theoretical analyses and retrievals on test cases," JQSRT, 189, doi:10.1016/j.jqsrt.2016.12.022 (2016) by Bauduin et al. MOPITT is also equipped with near-infrared channels which can boost surface-level sensitivity in some scenes.

p. 4, line 1. If scenes with low clouds are tolerated, what effect does that have on the retrieval vertical sensitivity (averaging kernels)?

p. 4, line 13. The assumption that all of the pyrogenic CO remains in the boundary layer is dubious. Can it be assumed that pyroconvection out of the boundary layer does not occur? After the first day or so of burning, it is likely that CO in the boundary layer will start venting into the free troposphere, thus affecting CO concentrations in the free troposphere throughout the region. Finally, there seems to be no consideration of the uncertainty of the boundary layer height.

p. 4, line 28. The evidence that light path lengthening (in the presence of smoke) is insignificant is not compelling. The most credible way to prove this claim would involve

validation results. If the evidence for this claim is based solely on retrieved amounts of methane and water vapor, those results should be presented in the manuscript to allow the reader to judge whether in fact retrievals of those gases 'are not considerably increased compared to the pre-fire background abundances.'

---

## Author Comment (AC1) · 12 Jul 2019

**Final response to referee comments on paper acp-2019-5**

First of all, we would like to thank both reviewers for their critical and constructive comments, which helped to significantly improve the manuscript. The concise letter-style of the manuscript was not fully adequate for the presentation of a new field of application of satellite data and was replaced by a more detailed edition with considerably extended analysis, discussion, and conclusions, in particular with respect to the associated uncertainties. We explicitly discuss and quantify uncertainties arising from boundary layer height, plume dynamics, and smoke aerosols in the revised version.

The data set providing the boundary layer heights was replaced by the ECMWF ERA5 reanalysis, which is available at hourly resolution. The previously used ECMWF analysis is only available at time steps of 6 hours (0, 6, 12, 18 UTC). Therefore, the maximum boundary layer height at local noon close to the time of the satellite overpass (21:30 UTC = 13:30 local time) was missed leading to underestimated heights. The usage of ERA5 provides far more realistic results. As a consequence, all city scenes, even the most polluted ones, likely comply with national ambient air quality standards, which is in line with isolated ground-based air quality measurements. The largest detected boundary layer concentration anomaly within all city radii (scene near Sacramento Airport on November 10) amounts now to $5.42\,\mathrm{mgCO\,m^{-3}}$ [3.97–5.96; $1\sigma$].

We also prepared the companion paper amt-2019-243 (Schneising et al., 2019), which describes the underlying algorithm in detail and includes error characteristics based on synthetic data, validation of the satellite data with reference data, and comparisons to the operational product.

Below we give answers and clarifications to all comments made by the referees (repeated in italics).

**Anonymous Referee #1**

**General comments**

**Reviewer:** *The style of the text is at the edge of what is acceptable for scientific writing. It uses emotional and judgemental wording, (non-exhaustive) list of examples: title: "devastating"; abstract: "one of the most disastrous months in Californian history", "destructive wildfires raging", "burnt to cinders"; introduction: "the town of Paradise was wiped out", "an unprecedented instance in history"; conclusion: "The analysed fires were the latest episodes of the deadliest and most destructive wildfire season the state of California has ever faced." Most of these statements can be removed without loss of any information.*

**Authors:** We agree with the reviewer and changed the style of the text at the passages in question but retained the description of the statistics of the California Department of Forestry and Fire Protection, e.g., that the wildfire season 2018 has been the most destructive on record with respect to burned land area, destroyed buildings, and fatalities.

**Reviewer:** *Further, the manuscript is very short in making reference to previous work. More references to earlier CO work of the MOPITT, SCIAMACHY, IASI, TES, AIRS teams are required.*

**Authors:** We extended the list of references considerably and give a more comprehensive review of earlier CO work in the revised version.

**Reviewer:** *What is the scientific value of the paper? The general CO detection capabilities of TROPOMI have been published before [e.g. Borsdorff et al., 2018a,b]. Air quality issues with wild-fire CO emissions are well-known. I would argue that the scientific value is the quantitative estimation of the CO burden (in units mg $m^{-3}$) based on daily recurrent satellite data i.e. the evaluation of TROPOMI's capabilities for dense CO-related air quality monitoring. Comparison to the CAMS model could also be an added value since it might trigger model improvement. Currently, the methodological evaluation and model comparisons are too short and too vague to serve any of these scientific purposes.*

**Authors:** We clarified that the main scientific value is indeed the dense daily recurrent satellite monitoring of the CO burden and extended the analysis and discussion of the methodology and the associated uncertainties to serve this interpretation. We also present the comparison to the CAMS model in more detail and discuss differences at a high resolution of $0.1° \times 0.1°$ and the potential of model improvement.

**Specific comments**

**Reviewer:** *P4, L11: Please add a discussion on errors coming from the assumptions on boundary layer height knowledge. Discuss how boundary layer height is determined. Please also add figures or tables for typical boundary layer heights. Are boundary layer heights of a few hundred meters (at midday) realistic (P7, L5)? These boundary layer heights need to be validated. If the boundary layer is so shallow, a large fraction of the fire emissions might reach above the boundary layer due to initial thermal rise.*

**Authors:** We now describe that ERA5 boundary layer heights are defined as the lowest height where the bulk Richardson number, which interrelates stability with vertical wind shear, reaches the critical value of 0.25. We also added a discussion of the uncertainty arising from boundary layer height including the inherent uncertainty estimate based on a 10-member 4D-Var ensemble and the temporal variation of the boundary layer height between 13:00 and 14:00 for a satellite overpass at 13:30 local time. Furthermore, we illustrate the diurnal variations of the boundary layer heights at the analysed cities and fires and compare them to IS4FIRES injection heights to evaluate if the fire emissions might reach above the boundary layer and to determine the uncertainty arising from plume dynamics. It is important to note that the hourly ERA5 boundary layer heights are considerably larger than the boundary layer heights derived in the previous version as the maximum at midday is better sampled due to the better temporal resolution.

**Reviewer:** *P4, L27: Please add and discuss a figure showing "the fact that the simultaneously retrieved gases, methane and water vapour, are not considerably increased compared to the pre-fire background abundances."*

**Authors:** In the original manuscript, it could be seen that this is true for the most polluted scene when comparing Figure 6 to Figure 7. In the revised version, we added a figure comprising maps showing the simultaneously retrieved methane abundances for all analysed days. On top of that, deviations of methane from the pre-fire background is also implemented as an alternative quality filter in the revised version because $XCH_4$ is far less variable than $XCO$ in the presence of wildfires and both gases typically exhibit similar error characteristics (Schneising et al., 2019). Hence, potential issues of the $XCO$ data, for example due to reduced near-surface sensitivity in the presence of clouds or smoke, are clearly detected in the corresponding $XCH_4$ data and filtered out. The figure also demonstrates that methane is not considerably increased compared to the pre-fire background abundances and that the $XCO$ enhancement patterns are not resembled in $XCH_4$. Thus, it can be excluded that the detected $XCO$ enhancement is only an artefact as a result of light path lengthening because of aerosol scattering at the particulate matter of the smoke, because such systematic errors would affect both retrieved gases similarly.

**Reviewer:** *P4, L30: The CAMS comparison is too short to be of scientific value. Please add a quantitative discussion (e.g. average TROPOMI on CAMS resolution and calculate departures).*

**Authors:** In the revised version, the CAMS near-real-time CO analysis is shown on a finer $0.1° \times 0.1°$ grid and an additional figure is added showing departures to TROPOMI, which is used to discuss the differences in more detail.

**Reviewer:** *P9, L24: None of the statements in the second paragraph of the conclusion are actually conclusions based on the scientific results of the paper, but rather they are author interpretation of climate change impacts.*

**Authors:** The revised version includes much more conclusions based on the scientific results of the paper, e.g., concerning TROPOMI's capabilities for dense air quality monitoring on a daily recurrent basis, the potential of model improvement, and the compliance with air quality standards. We highlight that the accurate determination of boundary layer concentrations depends on reliable external mixing layer height information and that the feasibility of the analysis is subject to specific favourable conditions affecting the vertical distribution of emissions in the case of fires to ensure that most of the fire emissions stay within the boundary layer and that pyro-convection or direct injection to the free troposphere is unlikely. Parts of the former second paragraph have been shifted to the introduction or removed.

**Reviewer:** *Figure 1: This figure could be dropped. The total column sensitivity of the solar absorption concept is standard scientific knowledge. Does the algorithm take into account that the near-ground sensitivity might be reduced due to scattering layers such as wildfire particulate plumes (or low clouds)? If not, what is the impact on the air quality derivations - does the satellite "see" the entire column?*

**Authors:** The sensitivity depends for example on the spectral resolution and the fitting window used and therefore potentially changes for different instruments or algorithms. Thus, quantitative details of the sensitivity are rather a feature of the instrument and algorithm than of the solar absorption concept. Therefore, it is important to show the averaging kernels. However, as the AKs are now shown in the companion algorithm paper, the figure is dropped

here. The newly implemented quality filter based on deviations of simultaneously measured methane from the pre-fire background ensures near-ground sensitivity for all scenes passing the filter. As a consequence, clouds and smoke near the origin of the fires are typically filtered out. However, in sufficient distance of the seat of fire the retrieved abundances are not affected and a quantitative analysis is still possible (e.g., in the analysed major cities), even in cases where efficient scattering in the visible spectral range is indicated by extensive plumes in the VIIRS images. The difference between scattering at clouds and wildfire smoke is the different particle size distribution leading to reduced visibility but far less scattering issues at the smaller particles of smoke in the far field of the fires in the $2.3\,\mu$m spectral range, where the satellite measurements are taken. This is supported by corresponding simulations included in the error analysis to quantify the impact of scattering at smoke aerosols.

**Reviewer:** *Figures 6 and 7: The figures are too dense with internal information. None of the acronyms (VC, SZA, VZA, dlnI/dx, ...) and few of the terms (sun-normalized radiance) are explained, the panels are too small, some of the panels are not even discussed (Temperature fit, ...). Recommendation: Either remove the figures entirely or just show the relevant parts e.g. the CO panels.*

**Authors:** Among other things, these figures should have demonstrated that the simultaneously retrieved gases are not considerably increased compared to the pre-fire background abundances, even for the most polluted scenes. As this was not realised by both reviewers, the figures are dropped and the mentioned fact is made more obvious by showing and discussing maps of methane and the newly implemented quality filter.

**Anonymous Referee #2**

**General comments**

**Reviewer:** *The first issue is the lack of any discussion of the retrieval algorithm or the expected error characteristics. References are provided to two ESA technical reports, but these appear not to be peer-reviewed, nor publicly available. A detailed presentation of the retrieval algorithm (in the peer-reviewed literature) is essential for establishing the provenance of the TROPOMI-WFMD CO products. While the general aspects of the algorithm might be similar to published algorithms developed for SCIAMACHY, some details will certainly be instrument-specific. Such details are extremely important.*

**Authors:** It is true that the two technical reports does not seem to be publicly available and we agree that a detailed presentation of the retrieval algorithm is essential. Therefore, we prepared the companion paper (Schneising et al., 2019) to fulfill this need.

**Reviewer:** *Similarly, details of the filtering methods (based on CO fit and water vapor absorption) will also influence the scientific interpretation of the data and should therefore also be discussed fully. (For example, how were the thresholds for CO fit and water vapor absorption determined?)*

**Authors:** The filtering method was updated in the revised version. The standard filter is

described and tested in Schneising et al. (2019). It is a machine learning approach trained globally based on cloud data from VIIRS and seems to be rather strict at least for California during the analysed time period, which is indicated by comparison with the VIIRS cloud product for days before the fire. Therefore, a new alternative quality filter for this local application based on deviations of methane from the pre-fire background is implemented and described in the revised version to get a somewhat larger amount of utilisable scenes but retaining good agreement with the VIIRS cloud product. The corresponding methane threshold (3 times the methane random error) was chosen to distinguish systematic from random deviations. As a consequence of the approach, potential issues of the XCO data, for example due to reduced near-surface sensitivity in the presence of clouds or smoke, are clearly detected in the similarly affected XCH$_4$ data and filtered out.

**Reviewer:** *The second major area of concern is the lack of any proper validation results; the only mention of validation is a reference to an unpublished technical report. The history of satellite remote sensing demonstrates clearly that satellite products can not simply be taken 'at face value'. For satellite CO products, validation should preferably exploit in-situ CO vertical profiles measured from aircraft. If that approach is not feasible for some reason, the authors could either exploit ground-based FTIR CO retrievals or other satellite CO products. (These latter methods are less optimal than in situ-based methods because of issues related to averaging kernels.) Comparisons of satellite CO products with surface measurements of CO concentration are generally inadequate for validation because of the variability of CO in the middle and upper troposphere.*

**Authors:** Proper validation with ground-based FTIR retrievals, which are in turn calibrated using in-situ aircraft measurements, is included in the companion paper (Schneising et al., 2019) showing that the TROPOMI/WFMD XCO data set is characterised by a random error (precision) of 5.1 ppb and a systematic error (relative accuracy) of 1.9 ppb. Thereby, averaging kernel issues are appropriately taken into account in the validation as documented in the companion paper.

**Reviewer:** *Beyond these two major issues, it is not clear how the TROPOMI-WFMD CO product relates to the TROPOMI-SICOR product (as developed by Borsdorff et al.). Do the two algorithms give the same results? Are there other issues which might make one product preferable over the other? Is the TROPOMI-WFMD CO product routinely generated and publicly available? These are inevitable questions that will concern potential users.*

**Authors:** The companion paper (Schneising et al., 2019) also includes comparisons to the operational product, which uses the SICOR algorithm concluding that both algorithms are highly correlated and show good global agreement although the algorithms differ in several respects. Thus, the scientific and operational products are predestined to be used together with other products in an ensemble approach to benefit from the large range of respective realisations of different physical aspects in the individual retrieval algorithms. This is discussed in the companion paper and is out of the scope of this paper. The TROPOMI/WFMD CO product is routinely generated and publicly available but with time delay.

**Reviewer:** *My advice to the authors is to strongly consider writing and submitting a validation paper (to AMT or another appropriate journal) which directly addresses these issues. Such a paper is an essential prerequisite to the quantitative use of satellite CO data. Publica-*

*tion of that paper would pave the way for this paper and increase its significance.*

**Authors:** The correponding paper addressing the raised issues is available for public review and discussion on AMTD.

**Specific comments**

**Reviewer:** *In several places, word choice could be improved to be less sensational and more scientific. For example, in the title, 'Devastating' could be 'Severe'. Similarly, the expression 'burnt to cinders' is gratuitous.*

**Authors:** We revised the style of the text at several passages and changed the title as suggested.

**Reviewer:** *p. 2, line 8. Need reference for physiological effects of CO on humans.*

**Authors:** We added a respective reference (Omaye, 2002).

**Reviewer:** *p. 2, line 15. The text in this paragraph suggests that MOPITT and IASI CO retrievals are generally insensitive to CO near the surface, but this is overly simplistic. In fact, publications document that both of these instruments can provide useful sensitivity to CO near the surface in daytime scenes over land (i.e., in conditions of high thermal contrast). For example, see "Sensitivity of MOPITT observations to carbon monoxide in the lower troposphere," JGR, 112, doi:10.1029/2007JD008929 (2007) by Deeter et al., and "IASI's sensitivity to near-surface carbon monoxide (CO): Theoretical analyses and retrievals on test cases," JQSRT, 189, doi:10.1016/j.jqsrt.2016.12.022 (2016) by Bauduin et al. MOPITT is also equipped with near-infrared channels which can boost surface-level sensitivity in some scenes.*

**Authors:** We rephrased the respective passage to avoid this potential misinterpretation and extended the list of references (including, e.g., Deeter et al. (2007), Bauduin et al. (2017), and Worden et al. (2010) for MOPITT's combined TIR/SWIR retrievals) to give a more comprehensive review of earlier CO work.

**Reviewer:** *p. 4, line 1. If scenes with low clouds are tolerated, what effect does that have on the retrieval vertical sensitivity (averaging kernels)?*

**Authors:** As already mentioned in the answers to the general comments, the filtering method was replaced in the revised version by a new alternative quality screening algorithn based on deviations of methane from the pre-fire background, which filters out scenes with reduced near-surface sensitivity in the presence of clouds or smoke by detecting significant underestimations in the simultaneously measured $XCH_4$ data. By construction, all measurements passing this quality filter are sensitive to CO near the surface because both gases, CO and $CH_4$, typically exhibit similar error characteristics (Schneising et al., 2019).

**Reviewer:** *p. 4, line 13. The assumption that all of the pyrogenic CO remains in the boundary layer is dubious. Can it be assumed that pyroconvection out of the boundary layer does not occur? After the first day or so of burning, it is likely that CO in the boundary*

*layer will start venting into the free troposphere, thus affecting CO concentrations in the free troposphere throughout the region. Finally, there seems to be no consideration of the uncertainty of the boundary layer height.*

**Authors:** In the revised version, we expanded the analysis, discussion, and conclusions, in particular with respect to the associated uncertainties. We explicitly added an estimation of the uncertainties of the determined concentration anomalies arising from boundary layer height and plume dynamics. The boundary layer heights are also compared to IS4FIRES injection heights to evaluate if the fire emissions might reach above the boundary layer and to determine the uncertainty arising from the vertical distribution of emissions near the source. To this end, we also discuss the local ambient atmospheric conditions (moderate to severe drought), which are favourable for dry smoke plumes being trapped in the boundary layer. We also note that there is no indication for Pyrocumulus or Pyrocumulonimbus in the VIIRS true color images. In summary, it is likely that most of the CO load stays within the boundary layer and that pyro-convection or direct injection to the free troposphere is negligible during the first days of the fire. However, partial venting to the free troposhere cannot be entirely excluded and it is concluded that unknown plume dynamics remains the largest source of uncertainty in the calculation of the boundary layer CO burden caused by wildfires.

**Reviewer:** *p. 4, line 28. The evidence that light path lengthening (in the presence of smoke) is insignificant is not compelling. The most credible way to prove this claim would involve validation results. If the evidence for this claim is based solely on retrieved amounts of methane and water vapor, those results should be presented in the manuscript to allow the reader to judge whether in fact retrievals of those gases 'are not considerably increased compared to the pre-fire background abundances.'*

**Authors:** In the original manuscript, it could be seen that the statement is true for the most polluted scene when comparing Figure 6 to Figure 7. In the revised version, we show this more comprehensively by adding a figure showing maps of methane for the analysed days demonstrating that methane is not considerably increased compared to the pre-fire background abundances and that the XCO enhancement patterns are not resembled in XCH4. Thus, it can be excluded that the detected XCO enhancement is only an artefact as a result of light path lengthening because of aerosol scattering at the particulate matter of the smoke, because such systematic errors would affect both retrieved gases similarly. In the error analysis, we also included simulations to quantify the impact of scattering at smoke aerosols.

**References**

Bauduin, S., Clarisse, L., Theunissen, M., George, M., Hurtmans, D., Clerbaux, C., and Coheur, P.-F.: IASI's sensitivity to near-surface carbon monoxide (CO): Theoretical analyses and retrievals on test cases, J. Quant. Spectrosc. Radiat. Transfer, 189, 428–440, https://doi.org/10.1016/j.jqsrt.2016.12.022, 2017.

Deeter, M. N., Edwards, D. P., Gille, J. C., and Drummond, J. R.: Sensitivity of MO-PITT observations to carbon monoxide in the lower troposphere, J. Geophys. Res., 112, https://doi.org/10.1029/2007JD008929, 2007.

Omaye, S. T.: Metabolic modulation of carbon monoxide toxicity, Toxicology, 180, 139–150, https://doi.org/10.1016/S0300-483X(02)00387-6, 2002.

Schneising, O., Buchwitz, M., Reuter, M., Bovensmann, H., Burrows, J. P., Borsdorff, T., Deutscher, N. M., Feist, D. G., Griffith, D. W. T., Hase, F., Hermans, C., Iraci, L. T., Kivi, R., Landgraf, J., Morino, I., Notholt, J., Petri, C., Pollard, D. F., Roche, S., Shiomi, K., Strong, K., Sussmann, R., Velazco, V. A., Warneke, T., and Wunch, D.: A scientific algorithm to simultaneously retrieve carbon monoxide and methane from TROPOMI onboard Sentinel-5 Precursor, Atmos. Meas. Tech. Discuss., https://doi.org/10.5194/amt-2019-243, in review, 2019.

Worden, H. M., Deeter, M. N., Edwards, D. P., Gille, J. C., Drummond, J. R., and Nédélec, P.: Observations of near-surface carbon monoxide from space using MOPITT multispectral retrievals, J. Geophys. Res., 115, D18314, https://doi.org/10.1029/2010JD014242, 2010.

---

## Referee Report (RR1)

Reviewer's Comments on "Severe Californian wildfires in November 2018 observed from space: the carbon monoxide perspective" by Schneising et al.

General Comments

This paper reports an analysis of TROPOMI CO observations of fire emissions over California during 2018. TROPOMI CO total column measurements are used to estimate CO concentrations in the boundary layer which are compared to air quality standards. Comparisons with the CAMS analysis, in which MOPITT and IASI CO data are assimilated, are also presented.

While this paper does not report anything fundamentally new, the results should interest many readers of ACP. The presented methods are generally reasonable with one exception. Because the TROPOMI data do not provide any information with respect to the vertical distribution of CO, a general assumption is made in the paper that the pyrogenic CO remains in the boundary layer. This assumption is the basis of the equation used to convert CO total column to CO boundary-layer concentrations, and is also important for the analysis of errors due to the radiative effects of smoke aerosol. This assumption may be reasonable near the source region at the very beginning of a fire event, but for later days CO concentrations in the free troposphere will grow as the result of boundary layer venting. Thus, for a particular TROPOMI observation, the partitioning of CO between the boundary layer and free troposphere will, to some extent, depend on the transport of CO emissions produced upwind. In fact, for regions far from the source regions, the CO enhancement in the free troposphere could greatly exceed the enhancement in the boundary layer. Thus, the method presented by the authors can really only provide an upper limit for boundary-layer CO concentration. The authors seem to acknowledge the effect of venting in Section 3.2, but propose without evidence that the associated uncertainty of this effect is 25%. With respect to errors associated with the 'shielding' effect of smoke aerosol (also in Section 3.2), the authors present results from a simulation in which the smoke aerosol was confined to the lowest 2 km or so. Thus, this analysis is also based on the premise that no venting takes place from the boundary layer to the free troposphere.

Fortunately, the authors should be able to remedy these problems with the paper using the CAMS analysis. Specifically, the CAMS model output (CO vertical profiles) should be analyzed to determine the expected spatial and temporal dependences of the partitioning of CO between the boundary layer and free troposphere. This analysis would lead to much more robust estimates of errors in boundary-layer CO concentrations derived from TROPOMI. Similarly, the simulation of errors due to the shielding effect of smoke aerosol should include a case study (guided by the CAMS output) representing a case where the free troposphere was significantly affected by upwind venting of the boundary layer.

Specific Comments

p. 1, l. 13. Replace 'in line' with 'consistent'

p. 1, l. 20. 'conflagration' does not seem like a scientific term

p. 2, l. 13. What is the exact scaling factor (or method) to convert between CO concentrations in ppm and mg/m3?

p. 2, l. 26. Replace 'Up to now' with 'Until now'

p. 3, l. 12.  Add 'measurements' after 'carbon monoxide'

p. 3, l. 18.  Please elaborate on validation results (e.g., what are dominant sources of random and systematic error)

p. 3, l. 29.  Clarify meaning of 'similar error characteristics'

p. 3, next-to-last paragraph.  Clarify how retrievals are performed over the ocean;  the SICOR retrieval algorithm requires the presence of clouds in such scenes.  Are retrievals over the ocean as reliable as retrievals over land?

p. 4, l. 9.  Equation 1 should be moved here from Section 3.2

p. 4, l. 19.  'for days' should be more specific

p. 6, l. 3.  'obviously and 'unambiguously' are redundant

p. 7, Fig. 3 (and Fig. 4)  Dotted areas in figure are not easily distinguished visually from non-dotted areas.  Consider either changing size of dots or using alternative color.

p. 8, l. 4.  If no reference is given, this equation needs more explanation (for example, what is the significance of the near-surface averaging kernel)

p. 9, paragraph beginning on l. 6.  See general comments above.

p. 10, l. 9.  replace 'entire' with 'the entire state of'

p. 11, l. 1.  Justify statement that the VIIRS images show no indication of pyrocumulus

p. 12, l. 1.  What are 'irradiances'?

p. 12, l. 9.  This is not a realistic worst case scenario, since it does not represent effects due to the vertical distribution of smoke aerosol (see General Comments).

p. 14, l. 33.  Replace 'it is the other way round' with 'the opposite is true'.

---

## Author Response (AR2)

Bremen, February 5, 2020

**Letter to the Editor of manuscript acp-2019-5 "Severe Californian wildfires in November 2018 observed from space: the carbon monoxide perspective"**

**Dear Editor,**

on behalf of all co-authors I have prepared this document, which provides the point-by-point responses to the reviews and a highlighting of all changes made in the revised manuscript.

All remaining concerns of the reviewers have been addressed. Concerning the accuracy of the ERA5 boundary layer height, we introduced an additional uncertainty associated to the boundary layer height estimation method based on the comparison of ERA5 PBL heights to boundary layer heights derived from lidar measurements. The question of PBL venting and transport to higher altitudes has been resolved by an analysis of CAMS CO vertical profiles providing a more realistic estimation of the uncertainty arising from unknown plume dynamics. All analyses (existing and new) indicate that most of the emitted CO stays within the boundary layer during the analysed period. As a consequence, the estimated total uncertainties have increased only slightly.

We hope that the revised manuscript meets the high standards of ACP and is now acceptable for final publication.

Best regards,

Oliver Schneising (corresponding author)

**Response to referee comments on the revised submission of the manuscript acp-2019-5**

We would like to thank both reviewers for their constructive comments, which helped to further improve the manuscript. Below we give answers and clarifications to all comments made by the referees (repeated in italics). All remaining concerns have been addressed in the rerevised manuscript. The main improvements are:

- Introduction of an additional uncertainty associated to the boundary layer height estimation method based on the comparison of ERA5 PBL heights to boundary layer heights derived from lidar measurements.
- A more realistic estimation of the uncertainty arising from unknown plume dynamics based on the analysis of CAMS CO vertical profiles.

**Anonymous Referee #1**

**Reviewer:** The paper by Schneising et al. has improved compared to the previous submission. While I rate the new version publishable in ACP with only minor modifications, I would like to note that the conclusions changed substantially between the initial submission and the revision, both qualitatively and quantitatively.

The new estimates for the urban CO burden are about a factor 2 (or more) lower than in the initial submission, which is largely due to the fact that the heights of the planetary boundary layer (PBL) were severely underestimated (wrong time of day). So, while air quality standards for "some neighborhoods [...] are likely exceeded" (initial submission, abstract) in the initial submission, "even the most polluted city scenes likely comply with the national ambient air quality standards" (revision, abstract) in the revised assessment. Recalling the sensational wording and the lack of methodological information in the initial submission, I would like to emphasize that methodological rigor should have priority over swiftness and sensation in scientific publications.

Authors: Agreed.

**Comments**

**Reviewer:** I am still not convinced that the PBL heights are as accurate as claimed by the manuscript. The error estimate is based on the ensemble spread of the ECMWF ERA-5 model, which is actually quite small. But there is no discussion of potential systematic model errors (e.g. one could compare various models, not only runs of the same model). Is there any validation study showing how ERA-5 PBL heights compare to measurements? How does CAMS PBL heights compare to ERA-5?

Authors: The PBL height uncertainty was based on the ensemble spread and the variation between 13:00 and 14:00. It is true that this error estimate neglected the uncertainty of the

PBL height estimation method and that the actual total uncertainty may be larger than the rather small ERA5 ensemble spread suggests. Although, there is no comprehensive validation study of ERA5 PBL heights available, we introduce an additional uncertainty associated to the PBL estimation method based on the comparison of ERA5 PBL heights to boundary layer heights derived from aerosol and turbulence detection lidar measurements by different retrieval methods presented by Wang et al. (2019), who find that ERA5 boundary layer heights around the satellite overpass time (13:30) are about 300 m smaller than the measurement-based boundary layer heights. Unfortunately, the CAMS PBL heights are not explicitly available to the user. However, the analysis of CAMS CO vertical profiles, which has been added in the rerevised version of the manuscript, also suggests that this additional uncertainty is quantitatively reasonable (see new Figure 8).

Reviewer: P3, 114: I guess, one should also mention the required topography input.

**Authors:** The corresponding information has been added: "Thereby, the ECMWF dry columns are corrected for the actual surface elevation of the individual TROPOMI measurements as determined from the Global Multi-resolution Terrain Elevation Data 2010 (GMTED2010, United States Geological Survey (2018)) inheriting the high spatial resolution of the satellite data."

**Reviewer:** P10, 14; Fig. 7: "As can be seen in Figure 7, the IS4FIRES injection heights corresponding to the top of the plume are equal or smaller than the respective maximum boundary layer height at the location of the fires, with the exception of the first day of the Camp Fire." Is this statement true? If I get Figure 7 correctly, the injection heights of the vicinity of Sacramento (dark purple). So, actually, the Camp Fire injection height always exceed the PBL heights substantially. For the Woolsey fire (pink horizontal bars), one would need to compare to Los Angeles (dark blue). There, the statement appears kind of true although injection heights seem to reach always up to the top of the PBL. In consequence, the rationale for the Camp Fire needs to be changed.

Authors: In the discussion of Figure 7 the injection heights (orange and pink horizontal bars) are compared to the PBL heights at the fire sources (orange and pink dotted lines) to assess if CO is injected to the free troposhere. Hence, the cited statement is true (all the more when considering the additional uncertainty associated to the estimation method of the boundary layer height newly introduced) and the rationale does not need to be changed. Another question is how much of the CO is still in the PBL in the more distant cities and how much may have been vented from the boundary layer to the free troposphere in the meantime (see also answers to Referee #2). This has been addressed by analysing CAMS CO vertical profiles at the analysed cities reinforcing the assumption that most of the emitted CO stays within the boundary layer even four days after ignition and at a greater distance from the fire sources.

**Reviewer:** P14, l10: Are the boundary layer concentrations quoted for CAMS calculated by using CAMS PBL heights or the ERA-5 PBL heights? If the latter, how would the numbers change if one uses CAMS PBL heights?

Authors: It has been clarified that the CAMS boundary layer concentrations are also calculated using the ERA5 PBL heights as CAMS PBL heights are not explicitly available to the user. However, the analysis of CAMS CO profiles (Figure 8) suggests that the ERA5 and CAMS boundary layer heights are likely consistent within their uncertainties after introduction of the additional uncertainty associated to the PBL estimation method. Although the CAMS concentration anomaly is almost twice as high as the satellite derived anomaly, the error bars of the CAMS and satellite derived concentration anomalies almost overlap due to the relatively large uncertainties arising from boundary layer height estimation and unknown plume dynamics. This discussion has been added in the rerevised manuscript:

"For the grid-box comprising the mentioned satellite scene, CAMS predicts [...] a boundary layer concentration anomaly of  $10.23 \text{ mgCO m}^{-3}$  [6.51–11.04;  $1\sigma$ ] using the ERA5 boundary layer height and the associated uncertainties. Although this is almost twice as high as the satellite derived concentration anomaly and potentially exceeds the national ambient air quality standards, the error bars of the CAMS and satellite derived concentration anomalies almost overlap due to the relatively large uncertainties arising from boundary layer height estimation and unknown plume dynamics."

**Anonymous Referee #2**

**General comments**

**Reviewer:** This paper reports an analysis of TROPOMI CO observations of fire emissions over California during 2018. TROPOMI CO total column measurements are used to estimate CO concentrations in the boundary layer which are compared to air quality standards. Comparisons with the CAMS analysis, in which MOPITT and IASI CO data are assimilated, are also presented.

While this paper does not report anything fundamentally new, the results should interest many readers of ACP. The presented methods are generally reasonable with one exception. Because the TROPOMI data do not provide any information with respect to the vertical distribution of CO, a general assumption is made in the paper that the pyrogenic CO remains in the boundary layer. This assumption is the basis of the equation used to convert CO total column to CO boundary-layer concentrations, and is also important for the analysis of errors due to the radiative effects of smoke aerosol. This assumption may be reasonable near the source region at the very beginning of a fire event, but for later days CO concentrations in the free troposphere will grow as the result of boundary layer venting. Thus, for a particular TROPOMI observation, the partitioning of CO between the boundary layer and free troposphere will, to some extent, depend on the transport of CO emissions produced upwind. In fact, for regions far from the source regions, the CO enhancement in the free troposphere could greatly exceed the enhancement in the boundary layer. Thus, the method presented by the authors can really only provide an upper limit for boundary-layer CO concentration. The authors seem to acknowledge the effect of venting in Section 3.2, but propose without evidence that the associated uncertainty of this effect is 25%. With respect to errors associated with the 'shielding' effect of smoke aerosol (also in Section 3.2), the authors present results from a simulation in which the smoke aerosol was confined to the lowest 2 km or so. Thus, this analysis is also based on the premise that no venting takes place from the boundary layer to the free troposphere.

Fortunately, the authors should be able to remedy these problems with the paper using the CAMS analysis. Specifically, the CAMS model output (CO vertical profiles) should be analyzed to determine the expected spatial and temporal dependences of the partitioning of CO between the boundary layer and free troposphere. This analysis would lead to much more robust estimates of errors in boundary-layer CO concentrations derived from TROPOMI. Similarly, the simulation of errors due to the shielding effect of smoke aerosol should include a case study (guided by the CAMS output) representing a case where the free troposphere was significantly affected by upwind venting of the boundary layer.

Authors: It is true that the assumption that the pyrogenic CO remains in the boundary layer is the basis of Equation 1. We had already presented several indications that this assumption is reasonable for the presented fires during the analysed period:

- The IS4FIRES injection heights corresponding to the top of the plume are equal or smaller than the respective maximum boundary layer height at the location of the fires.
- The local ambient atmospheric conditions (moderate to severe drought within the analysed time period) are favourable for dry smoke plumes being trapped in the boundary layer also rendering later deep moist convection with transport to the free troposphere during the first days of the fire at another location unlikely.
- There is no indication for Pyrocumulus or Pyrocumulonimbus in the VIIRS true color images as there is no obvious cloud formation over the fires.

The remaining uncertainty due to potential venting to the free troposphere was proposed to be 25%. It has to be pointed out that this uncertainty was not meant to be valid for all imaginable fires (as the CO enhancement in the free troposphere can indeed exceed the enhancement in the boundary layer for exceptional fires), but for the fires analysed in this manuscript. We agree that the 25% seem somewhat arbitrary and take up the proposal to estimate more realistic uncertainties based on the CAMS CO vertical profiles at the analysed cities. To this end, we determine the CO fraction above the upper bound of the ERA5 boundary layer height uncertainty range (including the newly introduced uncertainty associated to the boundary layer height estimation method). The CO fraction in the free troposphere does not grow significantly during the analysed period over the cities considered and we get an uncertainty arising from unknown plume dynamics of 30% for San Diego and 10% for the other cities. In particular, the CAMS CO profile analysis further reinforces the assumption that most of the emitted CO stays within the boundary layer even four days after ignition and at a greater distance from the fire sources, namely at the analysed cities.

The CAMS CO vertical profile analysis also confirms that the aerosol scenario used in the simulations is a well suited worst case scenario for the analysed fires (in sufficient distance from the seat of the fire, i.e for the cities in question, with low visibility but decreasing scattering issues at larger wavelengths). It has been clarified that this is not a general worst case scenario for all wildfires (see also answers to specific comments). That the estimated uncertainty is indeed an upper bound for this case study is also confirmed by the fact that methane is not considerably increased compared to the pre-fire background abundances: with an error of 5% or more the XCO enhancement patterns would be resembled in XCH4, which is not the case.

**Specific comments**

Reviewer: p. 1, l. 13. Replace 'in line' with 'consistent'

Authors: Done.

Reviewer: p. 1, l. 20. 'conflagration' does not seem like a scientific term

Authors: Has been replaced by "extensive fires".

**Reviewer:** p. 2, l. 13. What is the exact scaling factor (or method) to convert between CO concentrations in ppm and  $mg/m^3$ ?

Authors: The exact scaling factor is 1.164 computed for normal temperature and pressure (NTP, 1 atm,  $20^{\circ}$ C). This information has been added to the manuscript.

Reviewer: p. 2, l. 26. Replace 'Up to now' with 'Until now'

Authors: Done.

Reviewer: p. 3, l. 12. Add 'measurements' after 'carbon monoxide'

Authors: Done.

**Reviewer:** p. 3, l. 18. Please elaborate on validation results (e.g., what are dominant sources of random and systematic error)

Authors: Validation results are documented in Schneising et al. 2019. The following information has been added to the manuscript: "The retrieval error sources can be grouped into systematic and random error components. Systematic errors typically occur when the analysed scenes are not well characterised by the forward model, particularly in the presence of strong scatters under challenging conditions concerning measurement geometry and albedo. The random component is dominated by detector noise and pseudo-noise determined by specific atmospheric parameters or instrumental features."

Reviewer: p. 3, l. 29. Clarify meaning of 'similar error characteristics'

Authors: Has been clarified: "Furthermore, both gases typically exhibit similar error characteristics regarding sign and percentage magnitude of systematic errors (Schneising et al., 2019)."

**Reviewer:** p. 3, next-to-last paragraph. Clarify how retrievals are performed over the ocean; the SICOR retrieval algorithm requires the presence of clouds in such scenes. Are retrievals over the ocean as reliable as retrievals over land?

Authors: It has been clarified that cloudy scenes are typically removed and that qualityfiltered ocean retrievals are mainly limited to sun glint or glitter scenes as a consequence of the otherwise weak signal above water surfaces. As a result, there is significantly less coverage over the ocean compared to SICOR. The validation at island sites does not indicate specific problems above ocean. Reviewer: p. 4, l. 9. Equation 1 should be moved here from Section 3.2

Authors: The equation has been moved accordingly.

Reviewer: p. 4, l. 19. 'for days' should be more specific

Authors: It has been specified that smoke overcast large parts of the state for nearly two weeks.

Reviewer: p. 6, l. 3. 'obviously and 'unambiguously' are redundant

Authors: "obviously" has been deleted.

**Reviewer:** p. 7, Fig. 3 (and Fig. 4) Dotted areas in figure are not easily distinguished visually from non-dotted areas. Consider either changing size of dots or using alternative color.

Authors: The dots have been somewhat enlarged and framed in black for better visibility.

**Reviewer:** p. 8, l. 4. If no reference is given, this equation needs more explanation (for example, what is the significance of the near-surface averaging kernel)

Authors: A more detailed explanation of this equation has been added.

**Reviewer:** p. 9, paragraph beginning on l. 6. See general comments above.

Authors: See answer to general comments above.

Reviewer: p. 10, l. 9. replace 'entire' with 'the entire state of'

Authors: Done.

**Reviewer:** p. 11, l. 1. Justify statement that the VIIRS images show no indication of pyrocumulus

Authors: We think that this statement is justified because there are no obvious clouds on top of the smoke plumes at the fire sources in Figure 1. This has been made more clear in the manuscript. Pyrocumulus are expected to look more like in this NASA image from the Australian wildfires in December 2019 with cloud formation over the fires:

Reviewer: p. 12, l. 1. What are 'irradiances'?

Authors: Solar irradiances are used to compute the sun-normalised radiances from the nadir radiances. "radiances and irradiances" has been substituted by "sun-normalised radiances" in the manuscript for the sake of better comprehensibility.

**Reviewer:** p. 12, l. 9. This is not a realistic worst case scenario, since it does not represent effects due to the vertical distribution of smoke aerosol (see General Comments).

Authors: It has been clarified that this is not a general worst case scenario for all wildfires. However, it is considered a worst case scenario for the presented fires during the analysed period because all analyses, namely 1) IS4FIRES injection heights, 2) CAMS CO vertical profiles, 3) local ambient atmospheric conditions (favourable for dry smoke plumes being trapped in the PBL), 4) VIIRS true color images, equally indicate that the assumption that the bulk of additional CO from the fires stays in the boundary layer is true for this case study:

"The used aerosol scenario [...] is considered a realistic worst case scenario for the analysed fires because it is at the upper end of optical depths and at the lower end of Ångström exponents for typical fire aerosols (Eck et al., 2009). Furthermore, the corresponding aerosol profile is consistent with the previous results about the vertical distribution of the emitted species during the first four days of the fires. Thus, the Camp Fire and the Woolsey Fire very likely exhibit less scattering in the 2.3  $\mu$ m spectral range than our model scenario assumes at least during the period analysed."

Reviewer: p. 14, l. 33. Replace 'it is the other way round' with 'the opposite is true'.

Authors: Done.

**References**

[revised manuscript text omitted]

$$\Delta \rho_{bl} = \frac{E_{\rm CO}}{h_{bl}} = \frac{\Delta v_{\rm CO} \cdot M_{\rm CO}}{N_A \cdot A_{\rm CO} \cdot h_{bl}} \tag{1}$$

where  $\Delta v_{\rm CO}$  is the enhancement (in units of molecules per area) relative to the pre-fire background. The molar mass of carbon

- 25 monoxide  $M_{\rm CO} = 28 \,\mathrm{g} \,\mathrm{mol}^{-1}$  and the Avogadro constant  $N_A = 6.022 \cdot 10^{23} \,\mathrm{molec} \,\mathrm{mol}^{-1}$  are used to convert molecules per area to mass per area;  $A_{\rm CO} = 0.95 \pm 0.05$  is the dimensionless near-surface CO averaging kernel characterising the boundary layer sensitivity of the retrieval determined for appropriate conditions (solar zenith angle  $\in [50^\circ, 60^\circ]$ , albedo  $\in [0.1, 0.2]$ ). The boundary layer height determines the available volume for pollution dispersion and is thus a critical parameter for air quality assessment. The ERA5 boundary layer height is defined as the lowest height where the bulk Richardson number, which in-
- 30 terrelates stability with vertical wind shear, reaches the critical value of 0.25 (ECMWF, 2018). The associated corresponding uncertainty estimates are based on a 10-member 4D-Var ensemble. Furthermore, an additional uncertainty associated to the estimation method of the boundary layer height is introduced, which is derived from a comparison of ERA5 boundary layer heights to lidar measurements (Wang et al., 2019). The areal variation of this anomaly is determined from the standard de-